# Arctic sea ice is an important temporal sink and means of transport for microplastic

Ilka Peeken [1], Sebastian Primpke[1], Birte Beyer[1], Julia Gütermann[1], Christian Katlein[1], Thomas Krumpen[1], Melanie Bergmann [1], Laura Hehemann[1] & Gunnar Gerdts[1]

Microplastics (MP) are recognized as a growing environmental hazard and have been identified as far as the remote Polar Regions, with particularly high concentrations of microplastics in sea ice. Little is known regarding the horizontal variability of MP within sea ice and how the underlying water body affects MP composition during sea ice growth. Here we show that sea ice MP has no uniform polymer composition and that, depending on the growth region and drift paths of the sea ice, unique MP patterns can be observed in different sea ice horizons. Thus even in remote regions such as the Arctic Ocean, certain MP indicate the presence of localized sources. Increasing exploitation of Arctic resources will likely lead to a higher MP load in the Arctic sea ice and will enhance the release of MP in the areas of strong seasonal sea ice melt and the outflow gateways.

[1] Alfred-Wegener-Institut Helmholtz-Zentrum für Polar- und Meeresforschung, Am Handelshafen 12, Bremerhaven 27570, Germany. Correspondence and requests for materials should be addressed to I.P. (email: ilka.peeken@awi.de)

Marine debris is a growing environmental concern as recent reports indicate that increasing quantities of litter disperse into secluded environments, including Polar Regions[1–3] and the deep ocean floor[4]. Plastic accounts for 73% of marine debris globally[5], and it has been estimated that about 8 million tons of plastic move from land into the ocean each year[6]. However, only 1% of this has been accounted for in terms of small plastic debris[7], highlighting that some of the major sinks of oceanic plastic litter remains to be identified. The Arctic Ocean is now in a state of rapid transition that is best exemplified by the marked reduction in age, thickness and extent of the sea ice cover[8]. The European Arctic margin is influenced by drift ice formed on the Siberian shelves and carried to the Fram Strait via the Transpolar Drift[9]. In contrast, the Fram Strait is the gateway that transports warm Atlantic water, via the West Spitsbergen Current to the Central Arctic[10], containing an anthropogenic imprint[11]. It is well known that regions of the Arctic Ocean are highly polluted owing to local sources and long-range atmospheric input[12]. In this context sea ice has been identified early on as a major means of transport for various pollutants[13,14], with north and east Greenland as well as the Laptev Sea, being especially prone to contamination from several sources[15]. A useful method to study sea ice drift pattern is by using passive microwave satellite images combined with the motions of sea ice buoys[16,17], which highlight the role of sea ice, e.g. spreading oil spills[18]. Recent studies stress the changes caused by the shift to first year ice resulting in the tendency of sea ice floes to diverge from the main drift pattern[19] such as the Transpolar Drift, with complex effects on exchange processes of any contaminants between the exclusive economic zones (EEZ) of the various Arctic nations[20]. Despite the scant knowledge of Arctic ecosystems, the trend towards thinner sea ice and ice-free summers in the future has already stimulated increasing exploitation of its resources in terms of shipping, tourism, fisheries and hydrocarbon exploration[21].

Plastic degrades into smaller fragments under the influence of sunlight, temperature changes, mechanic abrasion and wave action[22]. Particles < 5 mm are called microplastics (MP) and have recently featured in a strongly growing number of studies and publications[23]. MP raise particular concerns because plastic in this size category can be taken up by a much wider range of organisms with currently largely unknown health effects on marine life and humans[24,25]. Over the past decade, MP were identified from numerous marine ecosystems globally[5], including the Arctic[2,26] and the Southern Ocean[3]. High concentrations of MP also occur in the surface waters south-west of Svalbard[26], but overall, MP origins, pathways and their compositional framework within Arctic sea ice remains unclear.

Here, we analyzed the content and composition of MP from sea ice cores at five different locations along the Transpolar Drift to assess if sea ice is a sink and transport vector of MP. Ice cores were taken from one land-locked and four drifting ice floes to distinguish between local entrainment of MP and long-distance transport. MP composition of the cores was analyzed by focal plane array detector-based micro-fourier-transform infrared imaging[27,28] (Imaging FTIR) and compared to a previous study with respect to MP in Arctic sea ice cores. Analyses of discrete ice core horizons allowed us to assess the spatial variability within sea ice and to reconstruct the location of MP incorporation. By computing drift trajectories, coupled to a thermodynamic ice growth model[29], possible source regions of MP entrainments during ice growth were identified.

## Results

**Microplastics in entire sea ice cores**. To quantify the MP concentration and composition we obtained ice cores during expeditions of the German research ice breaker *Polarstern* in spring 2014 and summer 2015 in the Fram Strait and Central Arctic (Fig. 1a, Table 1). The highest MP particle concentration ($(1.2 \pm 1.4) \times 10^7$ N m$^{-3}$) was detected in an ice core taken in the pack ice of Fram Strait (core B; Fig. 1b). The MP concentrations of all sea ice cores were highly variable with the second highest MP concentration found in the land-fast ice of the Fram Strait (core A; $(4.1 \pm 2.0) \times 10^6$ N m$^{-3}$). The MP load of core C and E, collected north of Svalbard and in the Nansen Basin respectively, varied between $(2.9 \pm 2.4) \times 10^6$ N m$^{-3}$ and $(2.4 \pm 1.0) \times 10^6$ N m$^{-3}$. The lowest concentration was found in core D $(1.1 \pm 0.8) \times 10^6$ N m$^{-3}$; Fig. 1b) from north of Svalbard. The values recorded in this study are two to three orders of magnitude higher than in a previous study from the Central Arctic[2] $(1.3–9.6) \times 10^4$ N m$^{-3}$, values exclude rayon, for further details see method section), which can largely be explained by the different methodology used. In the previous study[2], the filter area was first inspected by light microscopy and suspected MP particles were then analyzed individually by Fourier-transform infrared (FTIR) microscopy. In contrast, we used Imaging FTIR[27,28], where entire areas were scanned. This excluded the human bias introduced by visual selection of particles (Fig. 2a). Imaging FTIR includes the far more informative infrared region of the spectrum from the very onset of the analysis and enables the detection of very small particles (down to 11 μm), which are most likely overlooked by visual inspection and therefore not included in the majority of the previously published studies. By using this approach, we were able to show that most of the MP particles identified in the sea ice cores were smaller than 50 μm. On average 67% of the particles were within the currently smallest detectable size class of 11 μm (Fig. 3). Such small particles were not considered in the previous study on MP in sea ice by Obbard et al.[2]. Concerning the extremely high error values calculated for the individual samples it should be noted, that they result from the analyses of three different areas per filter indicating an unequal particle distribution (Fig. 2b). In contrast to our previous studies[30,31], the samples were not macerated beforehand and only $H_2O_2$ was applied after filtration to remove natural organic residues. MP particles tend to form hetero-aggregates with microalgae or natural organic matter[32] and therefore it is likely that these aggregates were printed on the membranes by filtration. Thereafter the organic matrix was removed by wet oxidation before the Imaging FTIR measurement. However, it should be noted that, to the best of our knowledge, our study is the first to indicate and document an uneven MP particle distribution on filters overall. With respect to the early days of direct bacterial counting, where similar observations were made, future MP studies should address this problem by improving the general sample preparation or by applying a statistically valid recording approach as used in direct bacterial counting[33]. We provide our measurements as particle count per volume for consistency with previous studies. However, we suggest that future studies also consider polymer-specific MP mass per volume data[34] to allow for calculation of fluxes or total load of synthetic polymers (independently of the degree of fragmentation).

In total, 17 different polymer types were identified (Supplementary Fig. 1; Supplementary Table 1), with polyethylene (PE), varnish (including polyurethanes and polyacrylates), polyamide (PA) also called nylon, ethylene vinyl acetate (EVA), cellulose acetate (CE alkylated), polyester (PES) and polypropylene (PP) contributing on average between 48% (PE) and 1.65% (PP) to the total measured MP composition. The following were on average below 1%: nitrile rubber, rubber, polystyrene (PS), polylactic acid (PCA), polyvinyl chloride (PVC), chlorinated polyethylene (PE-Cl), polycarbonate (PC), polycaprolactone (PCL), acrylonitrile butadiene (AB) and polyimide (PI) (summarized as others in

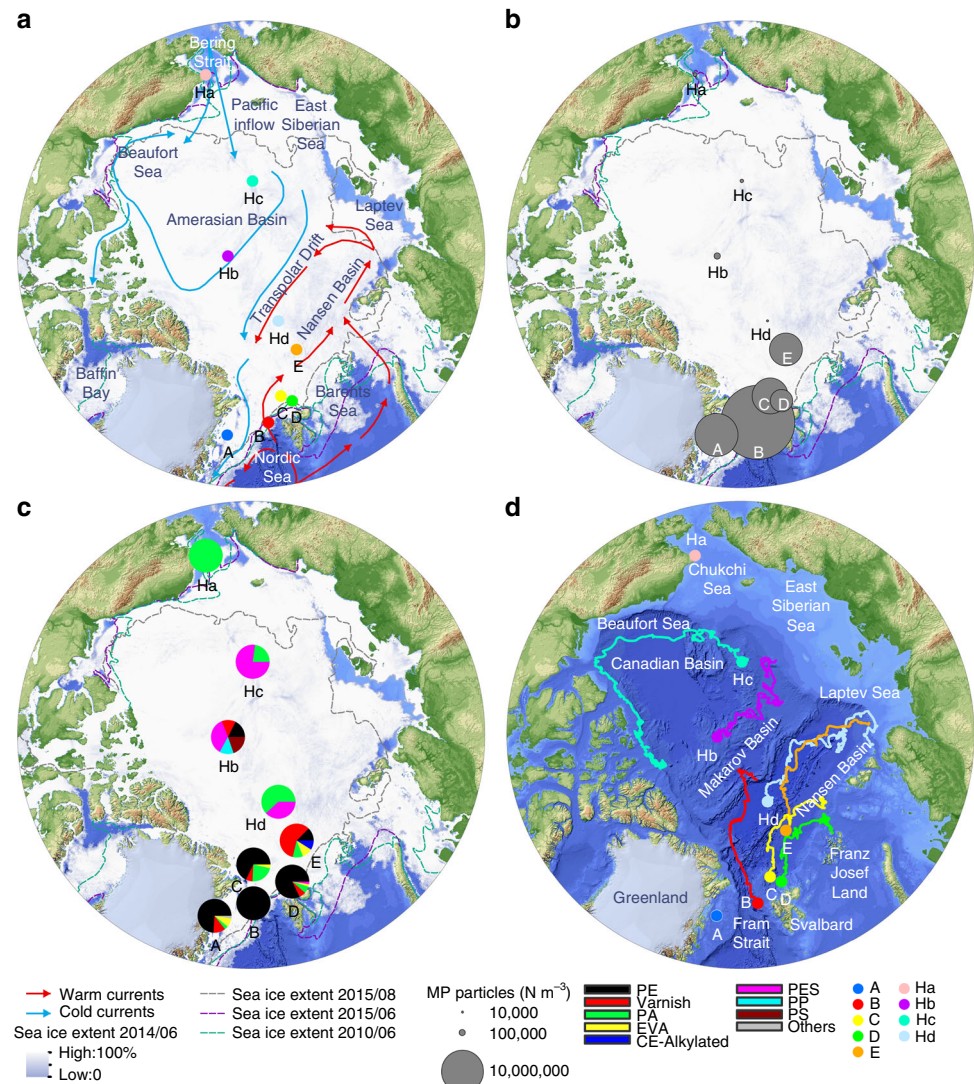

**Fig. 1** Pathway and microplastic content of sea ice cores in the Central Arctic. **a** Sampling position of sea ice cores (A–E) obtained during three Polarstern expeditions overlaid with the sea ice concentration (June 2014) and a schematic view of the major cold and warm water currents. Blue arrows indicate the inflow of Pacific water. For comparison, previously sampled sea ice cores[2] are included (Ha–Hd); **b** Total microplastic (MP) particle load m$^{-3}$ of the various sea ice cores (this study) and data reproduced from Fig. 2 of Obbard et al.[2]*; **c** Average % composition of polymers (polyethylene (PE), varnish (including polyurethanes and polyacrylates), polyamide (PA), ethylene vinyl acetate (EVA), cellulose acetate (CE-Alkylated), polyester (PES) and polypropylene (PP) and others) from the entire core (this study) and digitized data of figure two from Obbard et al.[2]*, acrylic equals varnish (others include acrylonitrile butadiene, chlorinated polyethylene, nitrile rubber, polycaprolactone, polycarbonate, polylactic acid, polyimide, polystyrene, polyvinyl chloride, rubber); **d** Drift trajectories of sea ice cores, except for land-fast ice station of Greenland (A) and the sample originating from the Chukchi Shelf Ha. The map was created using ArcGIS 10.3 and based on the General Bathymetric Chart of the Oceans (GEBCO)-08 grid, version 20100927, http://www.gebco.net, with permission from the British Oceanographic Data Centre (BODC). *The polymer rayon was excluded

**Table 1 Sea ice core sampling and accompanying information table**

| Core | Station ID | Sea ice type | Sample location | Sea ice origin | Campaign | Date | Latitude | Longitude |
|---|---|---|---|---|---|---|---|---|
| A | PS85_426 | Land-fast ice | Fram Strait | East Greenland | FRAM | 14 June 2014 | 78.27 | −14.71 |
| B | PS85_472 | Pack ice | Fram Strait | Makarov Basin | FRAM | 25 June 2014 | 79.75 | 4.30 |
| C | PS92_39 | Pack ice | North of Svalbard | Deeper Nansen Basin | TRANSSIZ | 11 June 2015 | 81.94 | 13.57 |
| D | PS92_32 | Pack ice | North of Svalbard | Franz Josef Land | TRANSSIZ | 06 June 2015 | 81.24 | 19.43 |
| E | PS94_54 | Pack ice | Nansen Basin | Laptev Sea | TRANSARC_II | 28 August 2015 | 85.09 | 42.61 |
| Ha | Site_12 | Land-fast ice | Chukchi Shelf | Chukchi Shelf | ICECAPE | 21 June 2010 | 68.30 | −166.98 |
| Hb | Site_H11 | Pack ice | Canadian Basin | Siberian Sea | HORTAX | 29 August 2005 | 84.31 | −149.06 |
| Hc | Site_H3 | Pack ice | Canadian Basin | Beaufort Gyre | HORTAX | 18 August 2005 | 78.29 | −176.68 |
| Hd | Site_H26 | Pack ice | Eurasian Basin | Laptev Sea | HORTAX | 19 September 2005 | 88.06 | 58.75 |

Please note the core labelling of sea ice cores for Fig. 1 from this study is A–E and from the study of Obbard et al.[2] Ha–Hd

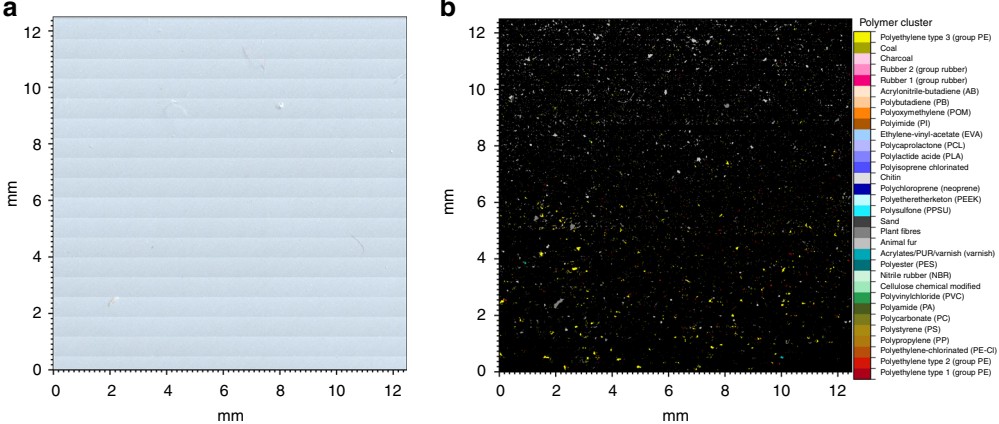

**Fig. 2** Images of the microplastic analysis. **a** Overview image collected by the fourier-transform infrared imaging (FTIR) microscope prior to measurement. **b** Polymer dependent false-colour image of an exemplary measurement field after FTIR measurement and automated analysis

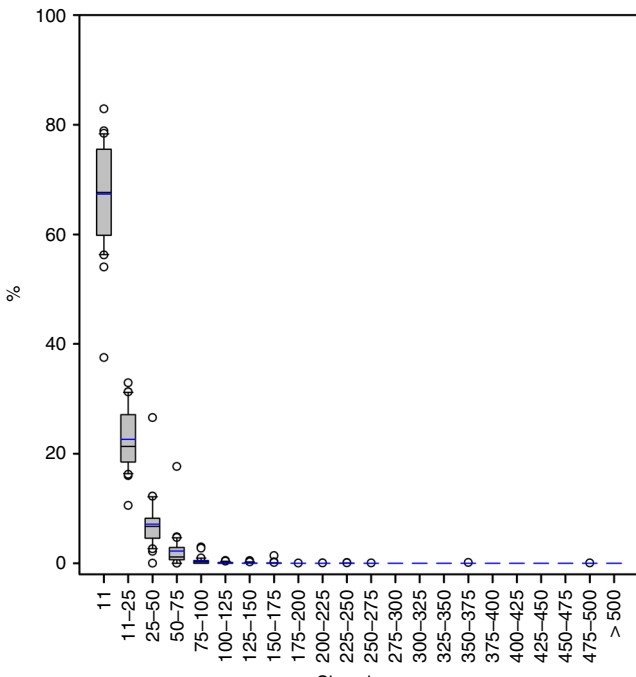

**Fig. 3** Size classes of observed microplastic particles. Box and whiskers plot of percentage (%) shares of MP numbers in different size classes in all sea ice cores. The boundary of the box closest to zero indicates the 25th percentile, the line within the box marks the median and the boundary of the box farthest from zero indicates the 75th percentile. Whiskers (error bars) above and below the box indicate the 90th and 10th percentiles. Blue lines indicate the mean and black circles indicate outliers

Figs. 1 and 4). Overall, the MP composition of the sampled sea ice cores was variable with PE being almost exclusively found in the upper horizons of core B sampled in the Fram Strait (above 90%; Fig. 1c). PE, which is among the economically most important polymers[35], also dominated the other core from the Fram Strait (A) and the cores retrieved north of Svalbard (core C, D; Fig. 1c). In these cores, PA, which is usually associated with fishing gear[35], accounted for 6 and 22% of the MPs. Both cores also contained varnish, which dominated core E taken in the Nansen Basin (Fig. 1c) and was present in the land-fast sea ice (core A, Fig. 1c). The polymer type varnish includes the previously described acrylic polymer type, known to account for up to 10% of the MP

in marine systems[35]. Core A and E also shared a relatively high proportion of EVA (up to 10%). The latter core was further characterized by almost 9% of CE-Alkylated (Fig. 1c), which is indicative of cigarette filters and commonly found in ocean debris[22]. Except for PE, overall the sea ice cores only partly reflect the composition of the globally produced polymers, which are dominated by PE, followed by PP, PVC, PS, PUR and polyethylene terephthalate (PET)[35].

**Sea ice trajectories and MP comparison to a previous study.** Large portions of sea ice are formed on the Siberian shelves[9]. Depending on individual ice floe drift patterns, they pass through different regions of the Central Arctic Ocean but are eventually carried to the Fram Strait via the Transpolar Drift[9,36]. To determine drift trajectories and source areas of sampled sea ice we tracked sampled sea ice backwards in time using low-resolution ice drift and concentration products from passive microwave satellites[17]. This back-tracking approach[17] showed that the sea ice samples originated from different source areas, namely the Amerasian and Eurasian Basins (Fig. 1d; Table 1). In particular core B can be retraced to the Makarov Basin, while the cores within the Eurasian Basin originated from the Laptev Sea (core E), near Franz Josef Land (core D) and the deeper Nansen Basin (core C). Except for the land-fast ice, east of Greenland (core A), all ice cores encountered the main path of the Transpolar Drift pack ice[9,36]. To enable a comparisons with Obbard et al.[2], we applied the back-tracking approach to the four sea ice cores from that study (Fig. 1d). The origin of the sea ice cores from the earlier study[2] only overlap in the region of the Laptev Sea (core Hd) with one of our cores (E), while the other cores can be related to the Beaufort Gyre (core Hc), the Chukchi Shelf (core Ha) and the East Siberian Sea (Fig. 1d). The strikingly high contribution of varnish (58%) in the sea ice cores originating from the Laptev Sea in our study (core E; Fig. 1c) did not feature at all in the earlier core Hd. However, even in the previous study, acrylic (similar type as varnish) was present in one core (Hb) and the contribution was in the same order of magnitude as described in other studies[35]. PA seemed to be a common compound observed in eight out of the nine investigated cores in both studies (Fig.1c) and was particularly dominant in the core from the Chukchi Shelf (core Ha). Ice cores analyzed by Obbard et al.[2] also showed high concentrations of PET (Fig. 1c), which might be partly reflected in our polymer type PES. The large difference between both studies might be related to our exclusion of fibres. Alternatively, it might reflect the increased effort to recycle this particular compound[37].

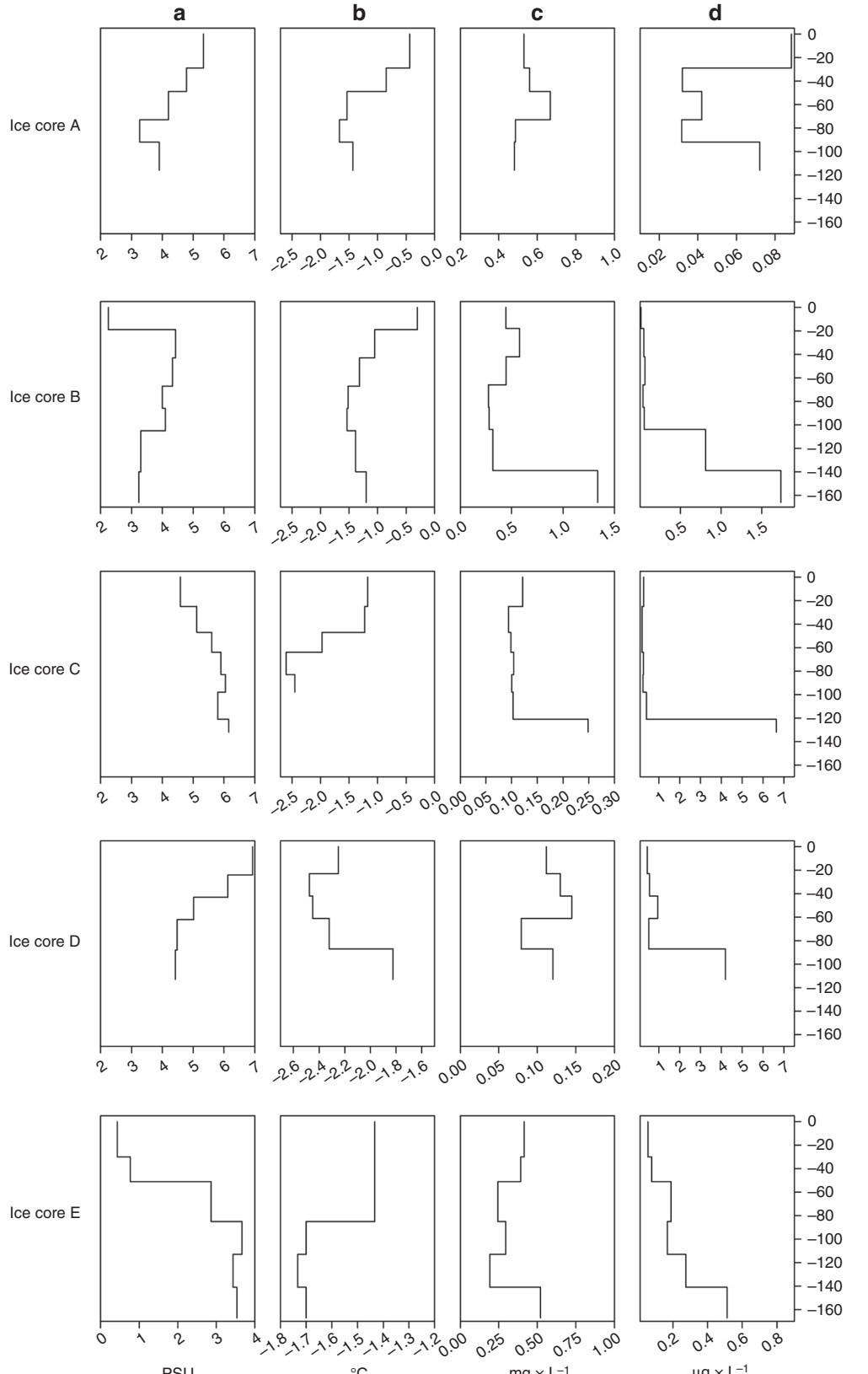

**Fig. 4** Vertical distribution of ancillary data in sea ice cores. **a** Refers to the salinity (PSU), **b** refers to temperature (°C), **c** refers to particulate organic carbon content (POC; mg×L$^{-1}$) and **d** refers to chlorophyll a concentration (µg×L$^{-1}$) for each core. Steps indicate the sampling horizons taken for each core

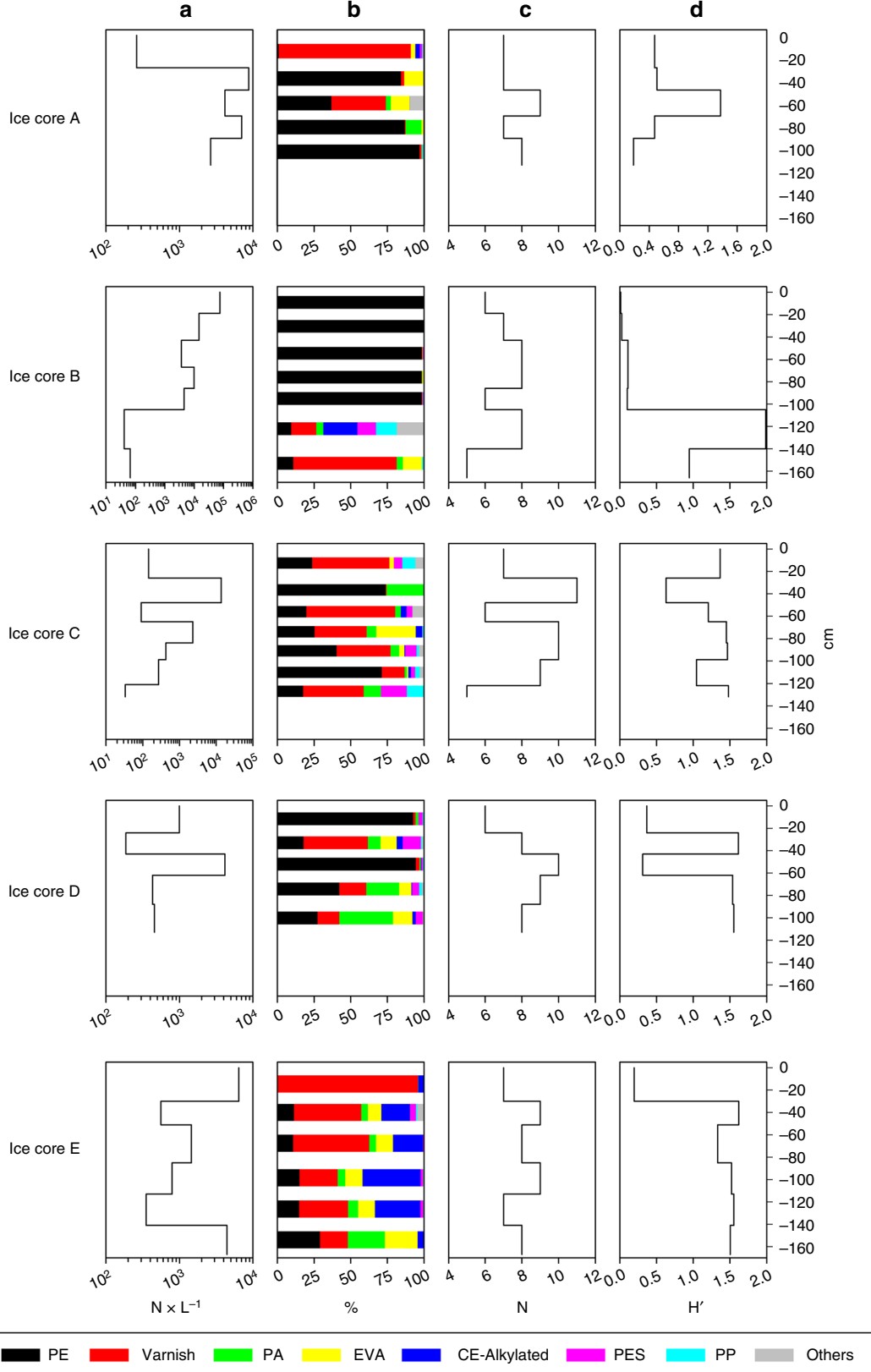

**Fig. 5** Vertical distribution of microplastic in sea ice cores. **a** refers to the concentration of microplastic particles (in N×L$^{-1}$) for each core. **b** refers to the polymer composition for each core: polyethylene (PE), varnish (including polyurethane and polyacrylate), polyamide (PA), ethylene vinyl acetate (EVA), cellulose acetate (CE-Alkylated), polyester (PES) and polypropylene (PP) and others. **c** refers to polymer richness (N), and **d** refers to the Shannon–Wiener index (H'). Steps indicate the sampling horizons taken for each core

**Ancillary variables and MP along sea ice cores**. Since the formation and growth of sea ice is a process in space and time, it is most likely, that the MP composition of the seawater–ice interface reflects the respective water body, which is in contact with the drifting sea ice floes as shown by the drifting path of the sampled ice floes (Fig. 1d; Supplementary Fig. 2). Given our current lack of knowledge about MP contamination of surface waters from the Central Arctic the mere consideration of the complete core is probably not sufficient with respect to the spatiotemporal history of the sea ice floe. In order to consider this spatiotemporal history we studied the vertical distribution of MP in the context of environmental and biological variables (ancillary variables) in each horizon (~20–30 cm sections) of the sea ice cores (Figs. 4 and 5). Temperature and salinity measurements suggest that the sampled sea ice consisted of first and second year ice (Fig. 4). Second year ice was indicated by a low salinity in the surface (core C & E). With a maximum of 1.67 m (core E), sea ice thickness was very low for spring conditions, but is in agreement with the observed reduction of sea ice in the Fram Strait[36] and Central Arctic[38]. The highest concentrations of ice algal biomass, as indicated by chlorophyll a, were mostly found in the lowest core horizons (Fig. 4; max. 6.64 µg L$^{-1}$), with a general L-shaped distribution pattern. Only the land-fast-ice core (A) had a maximum concentration at the top. Particulate organic carbon (POC) had a maximum of 1.34 mg L$^{-1}$ and also showed an L-shaped distribution pattern (Fig. 4). Only the land-fast ice core (A) and one core sampled north of Svalbard (D) had their POC maximum in the middle of the core.

In contrast to these generally L-shaped vertical profiles of biological variables, MP quantities in the various sea ice core horizons were extremely variable with concentrations ranging between 33 and 75,143 L$^{-1}$ (Fig. 5, units are adjusted to environmental data (L$^{-1}$)). The latter concentration was found in the surface horizon of core B originating from the Makarov Basin (Fig. 1d). Another high surface value with 6421 L$^{-1}$ was observed in core E, while all other cores contained relative low MP concentration in the top 20 cm (max. of 992 MP L$^{-1}$). The core originating from the Laptev Sea (E) displayed another MP maximum in the bottom horizon at the seawater–ice interface (4437 L$^{-1}$), while core C was characterized by a sub-surface maximum (Fig. 5). All other cores were characterized by variable concentrations reaching a maximum concentration of particles throughout the middle of the ice core, between 4159 and 13,794 L$^{-1}$ (Fig. 5). The synthetic polymer composition of the individual horizons showed strong differences within a single sea ice core (Fig. 5). PE dominated the synthetic polymers in the top 100 cm of the sea ice core B originating from the Makarov Basin (Fig. 5). PE was also present in all other cores but with highly variable contributions in the different horizons. In the core originating from the Makarov Basin (B), a change in salinity was associated with a drop in MP numbers and a shift to a more complex polymer composition, with high proportion of varnish in the bottom horizon (Figs. 4 and 5).

However, only few significant correlations between MP numbers, MP derived diversity indices (N, H', Fig. 5), MP sizes and ancillary variables were found (Supplementary Table 2). Highest Spearman rank correlations (negatively correlated) were found between chlorophyll a, total MP numbers, numbers of PE particles and the numbers of particles in the six lower size classes (covering the sizes 11–125 µm). This underscores that sea ice does not have a uniform MP imprint throughout the whole ice body and that the MP numbers or the polymer composition cannot be explained by ancillary variables recorded in parallel in the different horizons. This finding might point to a patchy distribution of MP in Arctic waters where local MP populations of the respective waterbodies are archived when in contact with

the ice–seawater interface. This is evident for sea ice cores C and D, which had extremely different polymer distributions even though both sampling locations were in close proximity north of Svalbard. Indeed, the back trajectories of these two cores revealed different sea ice origins and pathways along the Transpolar Drift (Fig. 1d). The core originating from Franz Josef Land (core D), showed two horizons dominated by PE polymers, with varnish and PA distributed along the core, particularly in the two lowest horizons. In contrast, the core originating from the deeper Nansen Basin (C) showed a strong variability of polymer distributions in all horizons with a higher proportion of PE and varnish and with the occurrence of PA, EVA and PES (Fig. 5). The highest contribution of varnish in all sea ice core horizons was evident in the core originating from the Laptev Sea (E). The middle part of this sea ice core was further characterized by various proportions of CE-Alkylated. This compound was also found in one horizon of core B, together with PP, which was also present in few horizons from core C (Fig. 5). Overall results from the 1D sea ice growth model highlight localized polymer entrainment (Supplementary Fig. 3), whereby, e.g., PE is present in high concentrations associated with the Atlantic and Pacific inflow in the Central Arctic, while e.g. varnish and EVA are more concentrated in the eastern region of the Eurasian basin (for details see Supplementary Fig. 3).

Cluster analyses of MP numbers, MP sizes and ancillary variables resulted in contrasting sample groups. No clear groups of the different cores or horizons were obtained for MP numbers and sizes (Supplementary Figs. 4, 5 and 6). In contrast, the ancillary variables grouped cores A, B and E and C and D (Supplementary Fig. 5). These results were also supported by the SIMPROF analysis. The clear separation found in the ancillary data might be related to the drift trajectories of the floes, where cores A, B and E follow the main path of the Transpolar Drift[9], while cores C and D drifted nearer the Atlantic water inflow[10]. Cores C & D (SIMPROF group c) displayed significantly higher Chl a concentrations, salinities and C/N-ratios compared to A, B and E (SIMPROF group d). Still, POC, PON and temperature were significantly higher in cores A, B and E (Supplementary Table 2). However, when superimposing the SIMPROF grouping of the ancillary variables on the polymer-specific MP numbers or MP derived diversity indices in the different core horizons, with the exception of nitrile rubber and PVC-particles there was no significant difference displayed in average between the two groups. This was also the fact for the MP numbers in all size classes (Supplementary Table 3 and Supplementary Data 1 and 2). Overall, it can be assumed that all environmental and biological variables are characterized by a strong seasonality[39,40], while MP particles once they are incorporated into the sea ice during sea ice growth seem to be more stationary in the ice core matrix.

## Discussion

Combination of Imaging FTIR[27] and an automated polymer identification approach[28] revealed that MP concentrations in Arctic sea ice are extremely high and therefore sea ice can be seen as a temporary sink for MP. However, even during winter large fractions of the sea ice are exported southward and eventually doomed to melt, with highest sea ice export fluxes out of the Fram Strait[17]. Hence, Arctic sea ice can be considered a temporary sink, a source and an important transport vector of MP, at least to the Fram Strait and North Atlantic.

The role of sea ice to redistribute, e.g. coastal sediments[41] and contaminants[15,42] along the Transpolar Drift or into the Central Arctic has long been recognized. It could be shown that the particular region of the Fram Strait will always be reached by any

contamination source from the distant Arctic. The estimated time between contamination and arrival in the Fram Strait ranges between two and four years for sources in the Laptev and Kara Seas, and up to six to eleven years from sources of the Amerasian Basin[15]. Although the recent circumpolar TaraOceans Expedition highlighted certain hotspots of floating plastics in the Arctic transported via the poleward branch of the Thermohaline Circulation[1], nothing is known to date about the actual MP concentrations in the waters of the Central Arctic Ocean itself, where all sea ice cores were collected.

Overall, the sources and sinks of MP are currently not very well understood[43], but since the Fram Strait is one of the main inflow gateways[44] to the Central Arctic Ocean, MP may have been transported with the relatively highly MP-contaminated offshore North Atlantic waters[45] into to the Arctic Ocean. For MP, this hypothesis was the focus of a recent study with a detailed sampling of (sub-) surface water south and west of Svalbard[26], but no clear transportation pathways of MP could be identified. In that study, water with a high melt fraction was rather low in MP particle load, suggesting a comparably low impact of sea ice originating MP. However, in our study we found several order of magnitude higher concentrations of MP in the sea ice and thus confirm the first study by Obbard et al.[2]. Particularly the abundance of the small particles (11 µm) are of concern, since they can be taken up within the microbial food web[25]. A recent modelling study, which suggested an active transport of plastic debris with the North Atlantic currents into the Central Arctic Ocean[46] corroborates the notion that polar waters can no longer be considered free of plastic litter. However, owing to the different analytical approaches applied in MP concentration estimates, a direct comparison of the studies is hampered by the current lack of standards in MP research and thus only general inherent trends can be compared. Terrestrial MP sources in these sparsely populated high-latitude regions seem to have a negligible contribution to the MP load of Arctic waters as Lusher et al.[26] reported extremely low MP concentrations close to shore. In contrast to other coastal (and more densely populated) areas, which are known to be much more contaminated with MP[24], emissions from Arctic terrestrial sources may be considered to be low. In addition, nothing is known about the impact of MP coming from the large rivers entering the Arctic. However, results from a recent modelling study suggest that contaminates originating from the Lena and Mackenzie river mouths, lead to a rapid spreading of theses pollutants in the entire Arctic Ocean[15].

Cores from east of Greenland (core A), close to Franz Josef Land (core D) and the Makarov Basin (core B) were further characterized by an abrupt change of MP composition in the various horizons. This suggests that the ice floe had passed through regions with fast changing MP compositions during the sea ice growth process. In analogy to previously described sediment entrainment into sea ice particularly frazil ice scavenged sediment particles[41]. During sea ice growth, characteristic salt fingers are developed[40] and enriching the particle concentration in the brine channels. In addition, coagulation with exopolymer particles excreted from sea ice algae[47] might further enrich the concentrations of MP. An accumulation of sediment particles in the surface horizons, as has been described for multiyear ice[41], due to constant surface melting, might explain the high MP concentration observed in the surface of the cores originating from the Makarov Basin and the Laptev Sea. It is however unlikely that this process redistributes MP in first year sea ice.

The highest concentration of MP ever determined in sea ice was found in core B, originating from the Makarov Basin. The core contained concentrations comparable to those from South Korean waters[48] or the Skagerrak[49], which are the highest hitherto reported values in terms of reports per volume unit

(www.litterbase.org). This peak consisted almost entirely of PE. Since PE has a low density[35], particles are likely to float over long distances at the sea surface with the ocean currents before they eventually sink through ballasting[50]. The Canadian Basin is supplied with water originating from the northeast Pacific and transported through the Bering Strait[51]. From the southern part of the Chukchi Sea sea ice has a direct path via the Central Arctic towards the Fram Strait[52]. We thus speculate that the high PE concentrations in the core from the Makarov Basin might reflect remains from the so-called North Pacific Garbage Patch[53], transported with the incoming Pacific inflow. A recent study by Desforges et al.[54] showed quite high MP concentrations for the NE Pacific and highlights the role of oceanographic conditions for the accumulation patterns of MP's[54]. Indeed, models project that on long time scales inter-ocean exchanges play a significant role in the distribution of marine debris enabling transport between accumulation areas[53]. Because of its widespread use and the low density, PE might nowadays, be considered to be a background MP even in surface water of the Central Arctic, analogous to the global distribution of certain persistent organic pollutants[55].

The high proportions of varnish are remarkable, particularly in the upper core from the land-fast ice (A) and the core originating from the Laptev Sea (E). In the latter, varnish is present throughout the entire core and often associated with a high proportion of cellulose acetate, indicative of cigarette filters[35] and to a lesser extent with ethylene vinyl acetate copolymer (EVA), a polymer, which is also used in antifouling paints for ships[56]. The occurrence of varnish and EVA can be attributed to ship traffic, which has increased between 2009 and 2014 in the Arctic[57,58]. PA, usually associated with fishing gear[35], was found frequently in almost all sea ice cores. This polymer was the most abundant in the sea ice core originating from the Chukchi Shelf[2]. It accounted for 22% in the core originating from the deeper Nansen Basin, which is comparable to other reports (<20%)[35]. Overall, a high contribution of PA may be related to increasing commercial fishing efforts in the eastern Bering Sea, Barents Sea, north of Svalbard and north of Franz Josef Land[58] implying local input. Fisheries accounts for a major share of the ship traffic in the Arctic Ocean[57,58] and continuous reduction of sea ice is assumed to increase fisheries further, particularly in the high Arctic[59]. The above attribution of source regions is supported by the reconstruction of the location of MP incorporation into the ice by coupling the back-tracking approach with a one-dimensional thermodynamic sea ice growth model (Supplementary Fig. 3).

Overall, this is the first detailed mapping of MP particle composition and size classes from sea ice cores obtained in the high Arctic with special emphasis on the vertical pattern of MP. The ice cores were characterized by various footprints of polymer composition resulting from different origins and pathways during the period of ice growth, although to prove this concept more data are needed. We identified unique footprints for different origin areas. This occurrence pattern is akin to data obtained for the presence of coloured dissolved organic matter[60], reflecting different source areas of the water, e.g., input from Lena river water or Pacific water. It is likely that parts of the MP, which are embedded in the sea ice, were transported by currents into these regions and that different oceanic realms (Pacific versus Atlantic) currently still have specific MP imprints. However, these imprints are altered by localized dispersal of MP in the Arctic, which need to be considered for future budgeting of global MP sources and sink estimates.

With respect to global climate change, large fractions of MP might be released from melting Arctic sea ice. Given a yearly melt of sea ice between $1.6 \times 10^4$ km$^3$ and $1.93 \times 10^4$ km$^3$ (PIOMAS 2011–2016 based on ref. [61]) large fractions of these particles are released. Basic calculations show the potential release of MP

between a minimum of $7.2 \times 10^{20}$ and a maximum of $8.7 \times 10^{20}$ particles per year between 2011 and 2016, assuming the here-observed average MP values. The maximum values can be attributed to the sea ice record minimum low found in 2012. Since currently only a few studies focus on the occurrence of MP in Arctic waters, it remains speculative whether these potentially released MP remain in Arctic waters or are transported to lower latitudes. On the other hand, due to the co-occurrence of sticky exopolymer particles in sea ice[62], a formation of hetero-aggregates might occur[32], resulting in a change of buoyancy of MP[63] and sedimentation to the seafloor. Indeed, very high numbers of MP were recently detected in deep-sea sediments of the HAUSGARTEN observatory in Fram Strait[30]. Highest MP concentrations occurred at the northern most stations, which are characterized by a long-lasting marginal sea ice zone area. Recent studies in the Central Arctic also showed that biogenic particles below 2 μm can contribute to the vertical flux[64] due to particle coagulation. The process of brine circulation due to melt progression[65] may redistribute MP in the Central Arctic Ocean as there is, for example, a high exchange of sea ice algae between the ice and the underlying water to depths of 40 m[66]. Many MP are in the same size range as sea ice algae, and may therefore also be transported far below the euphotic zone by brine convection.

We conclude that the MP distribution in the Central Arctic is more complex than previously considered, assuming only transport with high MP loads from the urban areas into the remote Polar Regions, although this undoubtedly constitutes the main point of entry. Our results also point to localized MP sources, which might become more pronounced as the exploitation of the Arctic progresses.

## Methods

**Sea ice coring.** Sea ice sampling has been carried out during three cruises with the ice breaker Polarstern in the Fram Strait (PS85, FRAM; June/July 2014[67]), the Barents Sea slope (PS92, TRANSSIZ; May/June 2015[68]) and the Central Arctic (PS94, TransArc II; August-October 2015[69]). At each station, a designated coring site was assigned and if present, the snow was removed before drilling the sea ice cores. Nitrile gloves were used and cores were drilled with a Kovacs 9 cm diameter corer (Kovacs Enterprise, Roseburg, USA). The microplastic cores were immediately transferred into plastic bags (polyethylene tube films (LDPE) by Rische and Herfurth) and stored at −20 °C.

**Back-tracking of sea ice.** To determine drift trajectories and source areas of sampled sea ice we tracked the sampled ice backward using low-resolution ice drift and concentration products from passive microwave satellites after Krumpen et al.[17] and Krumpen[29]. Sea ice concentration data used in this study were obtained from the National Snow and Ice Data Center (NSIDC). Ice drift data are provided by different institutions and have been widely used in various studies to investigate pathways and source areas of sea ice[17,70,71]. In this study, two different sets of ice drift products were used: During summer months (June–August), the Polar Pathfinder Sea Ice Motion product provided by the NSIDC provided on a 25 km grid[72] was applied. During the rest of the year, tracking is forced with sea ice motion data provided by the Center for Satellite Exploitation and Research (CERSAT) at the Institut Français de Recherche pour l'Exploitation de la Mer (IFREMER). Motion data are available with a grid size of 62.5 km, using time intervals of 3 days for the period between September and May[73].

The tracking algorithm works using motion and concentration data. A specific ice area is tracked backwards until: the ice reaches a position next to a coastline, the ice concentration at a specific location reaches a threshold value of <15% when ice parcels are considered lost, or the tracking time exceeds 4 years. To quantify uncertainties of estimated sea ice trajectories using satellite sea ice motion and concentration data, pathways of 39 buoys were re-tracked. Buoy data were obtained from the SeaIcePortal.de and followed from their deployment position in a forward direction. On average, the displacement of virtual buoys during the first 150 days (around 1000 km of ice drift) is around 35 km. After one year (ice drift of more than 2500 km), the average displacement is around 150 km. For general details of the AWI ICETrack tool (Antarctic and Arctic Sea Ice Monitoring and Tracking Tool), please see Krumpen[29].

**1D sea ice growth model.** Along the pathways derived from the back-tracking model sea ice trajectories, air temperatures (NCEP atmospheric reanalysis data) and snow depth (Warren climatology) and other atmospheric parameters were extracted as input for the thermodynamic ice growth model. The model was then used to estimate the location of MP incorporation into the respective ice sample (see below). To link the vertical distribution of MP within ice cores to the location, where MP particles were incorporated into the ice; we used a simplified one-dimensional thermodynamic model of sea ice growth. The model calculates sea ice growth based on surface air temperature, ocean heat flux and snow cover[74], where the change in ice thickness $\Delta h/\Delta t$ is given by:

$$\frac{\Delta h}{\Delta t} = \frac{-1}{L} \cdot (F_{\mathrm{OHF}} + (T_{\mathrm{surf}} - T_0)) \cdot \frac{\kappa_{\mathrm{i}} \cdot \kappa_{\mathrm{s}}}{(\kappa_{\mathrm{i}} z_{\mathrm{s}}) + (\kappa_{\mathrm{s}} z_{\mathrm{i}})} \quad (1)$$

where the latent heat of fusion $L$, the thermal conductivities of ice $\kappa_{\mathrm{i}}$ and snow $\kappa_{\mathrm{s}}$ and the freezing point of seawater $T_0$ are set to literature values according to[74]. Surface air temperature is extracted along the ice trajectory from NCEP reanalysis[75], while the ocean heat flux is assumed to be constant at 2 W/m$^2$ in agreement with Meyer et al.[76] and earlier work[77]. Ice thickness is calculated along the trajectories at daily increments and is in reasonably good agreement with ice core length for such a simple model. The model was also validated against an ice-mass-balance buoy[78] providing accuracy of few centimetres during the growth phase.

While such simple models perform very well on simple ice growth of typical Arctic sea ice, they are not suited to model strong regional melting features. Sampling locations of ice cores C and D, taken on the TRANSSIZ expedition in 2015, are strongly affected by small-scale abnormal basal melting in the inflow area of Atlantic water just north of Spitsbergen. Meyer et al.[76] report tenfold increased heat fluxes for the sampling region with basal melt rates of up to 26 cm/day. This dramatic melting, which is strongly limited in time and space can generally not be reproduced by single point to point comparisons even with the most sophisticated thermodynamic sea ice models. The difference between our simple model and measured ice core lengths, on the order of 50 cm, can be caused by just a few days of melting. This does, however, not affect the growth phase of the respective sea ice, so that our simple thermodynamic model can still be used to estimate the location of MP incorporation into the ice.

Using back-tracking (Supplementary Fig. 2) and the thermodynamic model we thus attributed a location to the incorporation of the different kinds of MP into the different ice samples, supporting our conclusions on the source regions of different MP species (Supplementary Fig. 3).

**Environmental and biological variables.** Handling the variables and measurements from sea ice cores was performed as described in previous studies[66,79].

**MP sample preparation.** To prevent contamination of samples, handling and processing the sea ice cores was conducted under a clean bench (Labogene Scanleaf Fortuna, Lynge, Denmark). Ice cores were cut individually into horizons ranging from 10 to 35 cm using a bone saw. To exclude sample contamination from sampling the surface of the ice core horizon was removed by a stainless steel grater. Afterwards, it was washed with 1 L of MilliQ to remove the particles that would eventually adhere. Each horizon was weighed before being melted in glass-preserving jars at room temperature and then concentrated onto Anodisc filters (47 mm, Whatman, Freiburg, Germany). All samples were treated with of 35% $H_2O_2$ (Roth, Karlsruhe, Germany, filtered over 0.2 μm Anodisc). After filtering the melt water the filter was overlaid with 40 mL $H_2O_2$ and incubated at room temperature overnight. Lastly, the $H_2O_2$ was drained and the filters were flushed with approx. 750 mL MilliQ water. To remove the adhering material, the filtration funnel was further flushed with 30% ethanol (VWR Chemicals, Darmstadt, Germany, filtered over 0.2 μm Anodisc) to reduce surface tension and thereby assure the concentration of all particles on the filter. The Anodisc filters were placed in glass petri dishes and dried at 30–40 °C in a drying cabinet (Memmert, Schwabach, Germany) overnight.

**Blank test.** For the blank tests, artificial ice cores were produced by freezing MilliQ water in a stainless steel beaker for one day. Afterwards, they were transferred into an ice core transportation bag and kept in the freezer for three days. Each day, the cores were rolled to simulate transport and then treated in the same manner as the sea ice core samples described above.

**Set up and operation of FTIR microscope (Imaging FTIR).** For the particle measurements, a Hyperion 3000 microscope (Bruker Optics) attached to a Tensor 27 (Bruker Optics) spectrometer was used. The microscope features a FPA detector with $64 \times 64$ detector elements. With a visual objective of ×4 magnification the whole filter area was photo-documented to obtain a sample overview. Afterwards, the IR measurement was performed via two 15× magnification Cassegrain lenses. The measurements and analyses were performed with the OPUS 7.5 software (Bruker). In previous studies, the optimal settings for the measurements were evaluated[27]. The scan was run in transmittance mode with 6 co–added scans, a range of 3600–1250 cm$^{-1}$ and a resolution of 8 cm$^{-1}$. A $4 \times 4$ binning was selected to balance the amount of data and the analysis time. The whole system was flushed with compressed dry air and with a flow rate of approx. 200 L h$^{-1}$ to prevent signals caused by air humidity and $CO_2$. After drying, the Anodisc filter was placed on the FTIR microscope. The background was measured on the Anodisc surface

without sample impurities. As it was not possible to analyze the whole filter cake of (36 mm) in diameter in a single measurement, three separate fields on the concentrated sample were measured as technical triplicates. Three grids of $70 \times 70$ FPA fields each were placed on the filter equalling 3,763,200 single spectra. The measurement took 12.25 h per grid resulting in a total analysis time of 36.75 h per filter.

**Quantification and identification method**. Each measurement field was subjected to the automated analysis by Primpke et al.[28]. During this process, each spectrum was compared twice against a spectral library with different data handling. Each successful hit was stored together with the $x$, $y$ and a quality factor into a csv file. Afterwards, the file was analyzed by image analysis to determine the polymer types, particle number per polymer and size distribution[28]. For each filter, the mean value $N_F$ and standard deviation of the three m technical triplicates were calculated. To extrapolate to the total area in contact with the sample, Eq. (2) was used:

$$N = \frac{N_F}{V_F} \quad (2)$$

The derived particle ($N$) numbers per litre melted ice were calculated from the mean value $N_F$ and the volume fraction of one measurement field $V_F$ from the overall volume. Particle numbers derived from blank samples were subtracted from $N_F$. To estimate errors of this conversion an error propagation was performed.

**Error propagation for calculation of particle numbers ($N$)**. To estimate errors for the technical triplicate, as well as the overall calculation and individual particle numbers ($N$) values, Eq. (3) was used:

$$\Delta N = \sqrt{\left(\frac{1}{V_F} \times \Delta N_F\right)^2 + \left(-\frac{N_F}{V_F^2} \times \Delta V_F\right)^2} \quad (3)$$

$\Delta N$ is the error of particle numbers per litre, $\Delta N_F$ is the standard deviation of the technical triplicate, and $\Delta V_F$ is the error of the conversion from full sample volume to volume per field. $\Delta V_F$ was calculated individually for each filter based on Eqs. (4) and (5)

$$V_F = \frac{V}{CF} \quad (4)$$

$$\Delta V_F = \sqrt{\left(\frac{1}{CF} \times \Delta V\right)^2 + \left(-\frac{V}{CF^2} \times \Delta CF\right)^2} \quad (5)$$

CF is the calculation factor derived from the measurement field size and concentrated filter area, and the sample volume of the investigated sample fraction. For $\Delta V$, an error of 0.01 L was estimated due to the gravimetric determination. For $\Delta CF$, an error value of 0.267 was found that includes the errors from the measurement field size determination and filter cake diameter.

**Quality factor thresholds for image analysis**. To maintain a 95% confidence interval for the chosen type of sample purification, several fields were manually reanalyzed[28]. For image analysis, the following quality factor thresholds differing from 600 were used (points with lower hit qualities were excluded from analysis): Polyethylene type $1 = 1100$, polyethylene type $2 = 1350$, PE-Cl $= 1310$, PC $= 700$, PA $= 1020$, PVC $= 800$, PES $= 800$, quartz $= 700$, EVA $= 900$, rubber type $1 = 1190$, rubber type $2 = 1300$, polyethylene type $3 = 1070$. In addition, polychloroprene had to be excluded due to the low hit qualities.

**Exclusion of rayon and fibres in this study**. The polymer **r**ayon is often included in the classification of MP found in the marine realm. For example, it accounted for up to 30% of the MP found in samples from the Arctic[2,26]. However, FTIR-based studies showed that 30% of suspected rayon fibres turned out to be cellulose, which is considered a natural product. Since cellulose and the semi-synthetic polymer rayon, have almost identical FTIR spectra[45], we resigned from identifying this particular compound. To compare ours with the previous study of MP in sea ice, we also excluded rayon from the MP identified in Obbard et al.[2]. We digitized the data from Fig. 2 (which according to the erratum are given as per liter) and up-scaled the numbers to N m$^{-3}$ for Fig. 1b.

The visible inspection classifies usually a large fraction of the MP as fibres[2,26], which cannot be identified with our way of applying the FTIR spectra, since fibres are not distributed flat on the surface of the filter and regions out of focus of the IR-beam were hardly or not detectable. The typical diameter of fibres was around 10–20 µm and the increased scattering due to their shape was not suitable for identification with our analytical approach.

**Statistical analyses**. Multivariate analyses were performed by using the software package Primer 7.012 (Primer-E), and univariate analyses by using Statistica 11 (Statsoft). Polymer-specific MP numbers and numbers of particles in the different

size classes were fourth root transformed; ancillary variables were normalized before multivariate analyses. For clustering (group average), Bray-Curtis similarities were used for MP-related data and Euclidean distances for ancillary variables. Calculation of diversity indices (richness, Shannon–Wiener H') were performed by using the PRIMER routine DIVERSE. The exploratory similarity profile test (SIMPROF) was applied to detect structures in the datasets. A significance level of 5% was used to test the SIMPROF statistic. SIMPROF groups (of ancillary variables; SIMPROF$^{av}$) were further used in ANOVAs for testing group differences of MP-related data and ancillary variables. Spearman rank correlations and ANOVAs were calculated by using non-transformed data.

**Data availability**. The authors confirm that all data underlying this study are fully available without restriction. All data can be downloaded from the public repository PANGAEA; https://doi.pangaea.de/10.1594/PANGAEA.886593.

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

## Acknowledgements

The TRANSSIZ cruise (Transitions in the Arctic Seasonal Sea Ice Zone; PS92; Grant No AWI_PS92_00) was initiated and co-organized by the Arctic in Rapid Transition (ART) network. The authors thank the crew of RV Polarstern and chief scientists Ingo Schewe (AWI_PS85_04) and Ursula Schauer (AWI_PS94_00) for their excellent support during this work. This study was funded by the PACES (Polar Regions and Coasts in a Changing Earth System) programme of the Helmholtz Association and contributes to the Pollution Observatory of the Helmholtz-funded programme FRAM (Frontiers in Arctic Marine Research). Sea ice pathways were calculated within the framework of the

Russian–German cooperation QUARCCS funded by the BMBF under grant 03F0777A. Furthermore, this work was supported by the German Federal Ministry of Education and Research (Project BASEMAN—Defining the baselines and standards for microplastics analyses in European waters; BMBF grant 03F0734A). This publication is Eprint ID 46090 of the Alfred-Wegener-Institut, Helmholtz-Zentrum für Polar-und Meeresforschung.

## Author contributions

I.P. and G.G. designed the study, J.G., B.B. and S.P. analyzed the MP in sea ice cores, T.K. calculated the sea ice origin, C.K. implemented the thermodynamic model and M.B. gave general advice about plastic litter. L.H. created the GIS figures. I.P. wrote the manuscript with contribution of all co-authors.

## Additional information

**Competing interests:** The authors declare no competing interests.

