## [Peer Review File · Nature Communications]

Reviewers' comments:

Reviewer #1 (Remarks to the Author):

The questions that this manuscript attempts to answer are interesting. However the samples studied do not allow for the comparisons and conclusions made. The manuscript is generally well written but in some places cited literature has been misrepresented.

My main concern is that the large variation of microplastics found in the ice cores and the small sample size precludes many of the analysis and comparisons made and certainly does not provide sufficient evidence of the effect of source area for the microplastics. This paper is an interesting documentation of the variation in microplastics seen and as microfibers were intentionally removed, may still present a lower estimate. These data are novel, but I suggest they are presented as observations of variation, rather than concluding too much from them.

I do appreciate the many challenges of getting ice samples, however with the variation in microplastic abundance and composition presented the results, further samples with replicates are required to understand the factors influencing the microplastic load in sea ice.

Reviewer #2 (Remarks to the Author):

Review of "Arctic sea ice is an important temporal sink and means of transport for microplastic" by Peeken et al.

The microplastic concentrations and composition in the sea-ice cores sampled in the Arctic Ocean are reported. Backtracking of sea ice was conducted to find the sources of microplastics within the sea-ice cores. The huge storage of microplastics in Arctic Sea ice potentially become a significant source of marine plastic pollution because of sea-ice melting in climate change.

This manuscript may become an important accomplishment in marine plastic pollution research. The survey results provided by the authors are quite shocking especially on surprisingly dense concentration of macroplastics in sea ice. If this is the case, sea ice will be indeed a serious threat causing marine plastic pollution in the future, as speculated by the authors. However, I recommend the authors to describe their results in a more careful manner; otherwise the readers feel their results less convincing. My comments follow.

[1] At this time, I have encountered some resistance to accept their surprisingly dense concentrations of microplastics. The concentration of $O(10^6)$ pieces/m³ (lines 54-56) is approximately four orders of magnitude higher than that in the sea ice sampled by Obbard et al. (2014); their estimate was 38-234 pieces/m³, which is NOT "one to two orders of magnitude higher" as described by the authors (line 57). In addition, $O(10^6)$ pieces/m³ is approximately six orders of magnitude higher than the pelagic microplastics in the upper Arctic Ocean (0.34 pieces/m³ provided by Lusher et al., 2015).

First, I questioned why the concentrations in sea ice are completely different between Obbard et al (2014) and the authors'. The authors mention the use of micro FTIR for that reason (line 61). If this is the case, the increase of the concentration was owing mainly to tiny microplastics <0.3 mm; usually

the lower limit of size in previous studies using conventional FTIRs. Please show the concentrations by each size bin especially in sizes smaller than several-hundred micro meters. The size composition of sampled microplastics will validate the authors' findings.

Second, I have a concern with the very large difference of microplastic concentrations in sea ice and upper ocean. Although Obbard et al (2015) also observed a difference between the abundances in sea ice and upper ocean, they could explain the difference in terms of scavenging because the POC level in sea ice is known to be the same order of magnitude higher than seawater. However, the huge difference (six orders of magnitude) "observed" in this study is unlikely to be explained by the scavenging phenomenon, and thus, the authors should give us alternative idea(s); otherwise the most plausible explanation for us would be the authors' overestimation of the microplastic concentrations in sea ice (see also my comment [10]).

[2] Backtracking of sea ice seems to be less convincing. My criticism is to track the 9-cm point (core diameter) in the map of 25-km and 62.5-km gridded data (Supplement, Method, Backtracking of sea ice). Apparently, such a small point cannot be resolved in these coarse gridded data. Please add the explanation how the authors trace 9-cm small point in the gridded map. Please validate this backtracking procedure. Please give us the error estimate.

[3] Please use the same "unit" in measuring microplastic concentration throughout the paper. The unit "m⁻³ (P.2)", "MP particles L-1 (P.4)", "MP L-1 (P.4)", "N/m³ (Fig. 1)", and "N L-1 (Fig. 2)" are used for the microplastic concentrations, which make scientists uncomfortable.

Specific points

[4] line 57, "one to two orders"; As mentioned in my comment [1], this should be "four orders".

[5] line 100-101, "MP composition...sea ice floes"; The polymer types detected by the authors are mostly heavier than seawater (1.025 g/cm³) except PP and PE. Such "heavy" pelagic plastic fragments have been actually observed in upper Arctic Ocean?

[6] line 123, "originating from the Canadian basin"; In my eyes, it was originating from the boundary between Canadian and Nansen Basins (Fig. 2d).

[7] lines 154-155, "It can thus be assumed... Atlantic water inflow"; I could not understand how the authors assumed the origins of MPs. The similarities of "biological and environmental parameters (line 151)" between A,B,E and Transpolar Drift, and between C&D and Atlantic water inflow are required.

[8] lines 214-215, "as those from South Korean waters...per volume unit"; These observations were conducted in an estuary and a lake, where microplastic concentrations are both completely different from those in open oceans such as the Arctic Ocean. Please refer Table 2 in Lusher et al. (2015), and Table 1 in Isobe et al. (2017), both cited by the authors.

[9] lines 219-221, “The high PE concentrations...incoming Pacific inflow”; This is unacceptable speculation. It is unlikely to have a direct debris path from the “North Pacific Garbage Patch” within the subtropical gyre to the Arctic Ocean, passing through the subarctic gyre in the North Pacific. Although the authors may misunderstand that the “North Pacific Garbage Patch” is the unique hotspot of microplastics in the North Pacific, this is not the case; please see Table 1 of Isobe et al (2017).

[10] lines 332-335, “the mean value of ...per litre melted ice”; One of my major concerns in reading this paper is here. It seems likely that they extrapolated from “measured areas” to “total area”, and converted from “total area” to “particle numbers”. The extrapolation and conversion need both careful validation and error estimates. Also, the detailed procedure from “area” to “number” should be mentioned here. What I concerned is the serious overestimation of particle numbers in converting the area measured using the micro FTIR.

[11] Figure S1(A and B); I am not familiar with nMDS analysis, and thus, please ignore this comment if this comes from my misunderstanding. Nonetheless, I have to state something that, in general, graphs without both the abscissa and ordinate are meaningless in sciences.

Reviewer #3 (Remarks to the Author):

Title: Arctic sea ice is an important temporal sink and means of transport for microplastic

Authors: Ilka Peeken, Sebastian Primpke , Birte Beyer, Julia Guetermann, Thomas Krumpfen, Melanie Bergmann, Laura Hehemann, Gunnar Gerdts.

In this paper, the authors are reporting on microplastic loads in ice cores collected in the Arctic Ocean in the spring 2014 and summer 2015 Polarstern cruises.

The data set is potentially interesting, but the paper is vague and the analysis of the data collected is not in depth. The paper simply describes the results without telling the reader a story from the data. The paper also does not cite relevant literature nor present the results in the context of prior work.

For these reasons, I recommend rejection of the paper in its present form.

Major Points:

- 1- The back trajectory model could have been used together with a simple 1D thermodynamic model to reconstruct the surface microplastic fields in the Arctic Ocean - as was done in Pfirman et al. (2004). This would add a new dimension and more depth to the paper.

- 2- The Arctic is presented by the authors as a pristine environment; yet the fact that pollutant are present in the Arctic is a known fact for a while – see for instance AMAP publications Arctic Monitoring and Assessment Programme, Pfirman et al., (1997), Rigor and Colony (1997), Korsnes et al. (2002), Pavlov (2007).
- 3- None of the Lagrangian Tracking literature is reviewed except for a recent paper by co-author Krumpfen et al.; see for instance, Tschudi et al., Fowler et al., Newton et al., Blanken et al., Szanyi et al (2016).
- 4- The concept that microplastic can accumulate at the surface because there is generally more ice growth than melt in the Arctic is not discussed. For instance, microplastic incorporated in sea ice at the base when ice freezes will migrate upward year after year and can accumulate at the surface. This is not mentioned in the discussion of the results and the large concentration observed at the surface in some cores; Wollenburg, 1993; Nuernberg et al., 1994.
- 5- The authors say the microplastic are conservative in places and in others they say that it can be lost through brine channels.

Reply to the reviewers:

Please note, reviewers' comments are marked in blue, while our answers are shown in black.

General remark to all reviewers:

We are very thankful for the detailed comments of all reviewers which were very helpful to improve our manuscript. Since the first submission of our paper, we achieved a major breakthrough in the automated analyses of MP particles published by Primpke et al. ¹. For example, in our previous analysis (based on Löder et al., 2015²) we were not able to provide MP particles sizes. Therefore, we now re-analyzed the entire data set by using a new, fully-automated analysis pipeline¹. The new method excludes any human error in identifying MP particles and with this we are able to provide more accurate and reliable data for identities, numbers and (now) also sizes. The relatively young field of MP research is rapidly evolving, but we are still far away from standard operational procedures. It is still difficult to compare the concentrations from the various published results and therefore data interpretation should still be done with caution. However, we are convinced that our analysis pipeline generates the most accurate data currently available for MP particles in the environment.

Please find below our detailed answer to the comments of the three reviewer.

Reviewers' comments:

Reviewer #1 (Remarks to the Author):

The questions that this manuscript attempts to answer are interesting. However the samples studied do not allow for the comparisons and conclusions made. The manuscript is generally well written but in some places cited literature has been misrepresented.

As mentioned above, we thoroughly revised the manuscript and reduced the over interpretation of our limited data set. We hope we were able to correct any misinterpretation of the cited literature, but since the reviewer did not explicitly mention which papers were misrepresented, we hope we cited the literature now in an appropriate manner.

My main concern is that the large variation of microplastics found in the ice cores and the small sample size precludes many of the analysis and comparisons made and certainly does not provide sufficient evidence of the effect of source area for the microplastics.

The new method applied allows for comprehensive mapping of filter areas at a resolution of ~10 µm. Most previously published studies relied, at best, on visual inspection and discrete identification of single particles by FTIR or Raman microscopy (only very few studies applied an imaging approach; Tagg et al., 2015³). Please keep in mind that so far nobody has performed this type of study before, in sectioning the

ice cores, and admittedly, we were also puzzled by our results. This is why we decided to apply the backtracking approach to look for different potential sources of MP particles during sea ice growth that could explain this variability. We agree that given the small number of cores studied, we cannot claim the effect of the source area and rewrote the entire manuscript to put less emphasis on this point.

For example L 282

“Overall, this is the first detailed mapping of MP particle composition and size classes from sea ice cores obtained in the high Arctic with special emphasis on the vertical pattern of MP. The ice cores were characterized by extremely different footprints of polymer distributions probably as a result of the origins and pathways during their development phases, although to prove this concept more data are needed “

This paper is an interesting documentation of the variation in microplastics seen and as microfibers were intentionally removed, may still present a lower estimate. These data are novel, but I suggest they are presented as observations of variation, rather than concluding too much from them.

As stated above, we took this concern very seriously and changed the focus in the revised version to the observation of variation, with the data closer related to the context of previous contamination literature available for the Arctic (The complete revision in responding to the comments can be followed in the annotated version of the original manuscript) As stated above, we clearly point out that more studies are needed to clarify our first observations (s. above):

I do appreciate the many challenges of getting ice samples, however with the variation in microplastic abundance and composition presented the results, further samples with replicates are required to understand the factors influencing the microplastic load in sea ice.

We thank the reviewer for acknowledging the challenges and we completely agree that, overall, more coherent data and experiments are required to fully understand all factors influencing the microplastic loads in sea ice. However, due to such strong differences between the various samples and, although agree it needs to be formulated carefully, we also feel it is necessary to look for possible explanations to stimulate future research. E.g. the upcoming one year-long drift study in the Central Arctic (MOSAIC) will allow for a much more detailed study on how MP are incorporated into the ice and how different oceanic currents will impact MP in sea ice. However, such ideal conditions to study these issues are very rare.

Reviewer #2 (Remarks to the Author):

Review of “Arctic sea ice is an important temporal sink and means of transport for microplastic” by Peeken et al.

The microplastic concentrations and composition in the sea-ice cores sampled in the Arctic Ocean are reported. Backtracking of sea ice was conducted to find the sources of microplastics within the sea-ice cores. The huge storage of microplastics in Arctic Sea ice potentially become a significant source of marine plastic pollution because of sea-ice melting in climate change.

This manuscript may become an important accomplishment in marine plastic pollution research.

We thank the reviewer for the overall positive evaluation of our work.

The survey results provided by the authors are quite shocking especially on surprisingly dense concentration of macroplastics in sea ice.

We think this is a misunderstanding, since we only report on microplastics. However, a recent study by Cozar et al. ⁴ showed also a high accumulation of macro plastics on the surface of sea ice in regions of the Central Arctic Ocean; and once they degrade they might fuel the microplastics incorporated into the drifting sea ice. This clearly shows that more studies are needed, particularly in the Central Arctic.

If this is the case, sea ice will be indeed a serious threat causing marine plastic pollution in the future, as speculated by the authors. However, I recommend the authors to describe their results in a more careful manner; otherwise the readers feel their results less convincing. My comments follow.

We generally agree with this concern and put our emphasis on carefully revising the manuscript, which can be followed in the annotated version of the original manuscript.

[1] At this time, I have encountered some resistance to accept their surprisingly dense concentrations of microplastics. The concentration of $O(10^6)$ pieces/m³ (lines 54-56) is approximately four orders of magnitude higher than that in the sea ice sampled by Obbard et al. (2014); their estimate was 38-234 pieces/m³, which is NOT “one to two orders of magnitude higher” as described by the authors (line 57).

Please note referring to the Obbard et al. paper⁵, there was a typo, which was corrected afterwards. I, therefore, copied the *Erratum* in italic below:

“In the originally published version of this article, the caption for Figure 2 contained a typo, which caused errors in the abstract and in section 4. The following have since been corrected, and this version may be considered the authoritative version of record.

In the abstract, “concentrations of microplastics at least two orders of magnitude greater than” was changed to “concentrations of microplastics are several orders of magnitude greater than...”

In the caption for Figure 2, “per cubic meter” was changed to “per liter.”

In Section 4, “microplastic concentrations we found are at least two orders of magnitude greater than” was changed to “microplastic concentrations we found are several orders of magnitude greater than...”

However, I agree it seems still unclear, since in the current available pdf version you still read at the beginning of the discussion the following: “38-234 pieces/m³”, which does not concur with the unit

change mentioned for figure 2. I contacted the author about this issue, but unfortunately never got any answer. Since we used the Figure 2 of Obbard et al.⁵ for the estimation of the particles and in the Erratum it is clearly stated that the data are “per liter”, we applied the correct order of magnitude, which was between 12805-96341 pieces m⁻³ **without the polymer rayon**. So overall our expression of 1-2 orders of magnitude was correct.

However, given our new calculations we need to change the value to 2-3 orders of magnitude (s. L65 “The values recorded in this study are two to three orders of magnitude”

As can be seen in the text, compared to Obbard et al.⁵ our method captures even the small particle fraction explaining the large differences between the two studies. Furthermore, in the study of Obbard et al.⁵, the samples were inspected visually by microscopy and only “suspicious” particles were identified by FTIR. This bears the risk the less suspicious, smaller particles might be overlooked. In contrast, our approach does not rely on an initial assessment (or classification; suspicious vs. non suspicious) of particles analyzed by FTIR, because we simply analyze complete filter areas and, therefore, excludes human bias.

In addition, O(10⁶) pieces/m³ is approximately six orders of magnitude higher than the pelagic microplastics in the upper Arctic Ocean (0.34 pieces/m³ provided by Lusher et al., 2015).

Again, please bear in mind that completely different methods were used in Lusher et al and our study. The data presented in Lusher et al. again rely entirely on the visible identification of MP particles. Only representative pieces were subjected to Fourier Transform Infrared spectroscopy (FT-IR) analysis to identify the MP composition. As we show in Figure 2 it is difficult to rely only on the visible inspection of particles. Thus, we feel a comparison of the actual numbers, which are based on these different methods, remains difficult; so studies are already underway on our side in surface samples using the same method. In addition, particles tend to concentrate in sea ice compared with ambient sea water during ice formation⁶ leading to ‘naturally’ elevated particle concentrations. Please see L241

“In analogy to previously described sediment entrainment into sea ice, particular frazil and anchor ice formations were responsible for scavenging of e.g. sediment particles⁴⁰. Overall, it is likely that during sea ice growth characteristic salt fingers are developed³⁹ enriching the particle concentration in the brine channels by possibly coagulating with exopolymer particles excreted from sea ice algae⁴⁶.”

First, I questioned why the concentrations in sea ice are completely different between Obbard et al (2014) and the authors’.

Please see the answer to comment 1.

The authors mention the use of micro FTIR for that reason (line 61). If this is the case, the increase of the concentration was owing mainly to tiny microplastics <0.3 mm; usually the lower limit of size in previous studies using conventional FTIRs. Please show the concentrations by each size bin especially in sizes smaller than several-hundred micro meters. The size composition of sampled microplastics will validate the authors founding.

We agree with the reviewer that it is important for the comparison to analyze the size ranges of the observed particles in order to explain the differences observed. By re-analyzing the entire data set we were now able to provide the requested size range, shown in the newly added Figure 3 in the results within the paragraph Microplastics in entire sea ice cores, and also adjusted the text adequately in L71:

“Imaging FTIR includes the far more informative infrared region of the spectrum already from the very onset of the analysis and enables the detection of very small particles (down to 11µm) which are most likely overlooked by visual inspection and therefore not included in the majority of the previously published studies. By using this approach we were able to show that most of the MP particles identified in the sea ice cores smaller than 50µm. On average 67% of the particles were within the currently smallest detectable size class of 11µm (Fig. 3). Such small particles were not considered in the previous study on MP in sea ice by Obbard et al.².”

Second, I have a concern with the very large difference of microplastic concentrations in sea ice and upper ocean. Although Obbard et al (2015) also observed a difference between the abundances in sea ice and upper ocean, they could explain the difference in terms of scavenging because the POC level in sea ice is known to be the same order of magnitude higher than seawater. However, the huge difference (six orders of magnitude) “observed” in this study is unlikely to be explained by the scavenging phenomenon, and thus, the authors should give us alternative idea(s); otherwise the most plausible explanation for us would be the authors’ overestimation of the microplastic concentrations in sea ice (see also my comment [10]).

Since no comparable study is currently available, which has the same details of MP identification; it is difficult to answer this concern. However, we would like to point out that although Obbard et al. used a totally different method, not accounting for the very small particles identified during this study, they suggested that scavenging was *also “several orders of magnitude higher in sea ice”* compared to the surrounding water (see erratum above). Thus, we have entrainment processes in sea ice, which currently are not very well understood. But, it is very likely that the sticky exopolymer particles, which are excreted from the algae, play a crucial role in entrapping the polymer particles from the water column during the sea ice growth. Certainly, we can exclude an overestimation of the MP particles as indicated in the detailed answer to comment 10.

[2] Backtracking of sea ice seems to be less convincing. My criticism is to track the 9-cm point (core diameter) in the map of 25-km and 62.5-km gridded data (Supplement, Method, Backtracking of sea ice). Apparently, such a small point cannot be resolved in these coarse gridded data. Please add the explanation how the authors trace 9-cm small point in the gridded map. Please validate this backtracking procedure. Please give us the error estimate.

Sea ice backtracking is a common method applied in sea-ice and related research. Tracking sea ice parcels is based on sea ice motion and concentration data from satellites. Since these are mostly passive microwave satellites with coarse footprints, the resolution of the derived motion fields is 62.25 x 62.25 km. The algorithm that is used to derive sea ice motion is based upon techniques like Maximum Cross-Correlation described in Emery et al.⁷. Fowler⁸ compared 10 x 10 pixel rectangular subsets of the same spatial locations between two consecutive days, and chose the location with the best correlation coefficient. The change in location is considered the ice displacement, which allows ice motion to be calculated. Hence, we do not track individual small points, but the displacement of rather large areas. The tracking technique is a standard application to identify source areas and pathways of sea ice and was used in several AWI publications explaining the spawning grounds of polar cod⁹, estimation the role of sea ice for the species distribution in sea ice¹⁰, as well as for the study of ice volume transport¹¹ in relation to litter on the seafloor of HAUSGARTEN observatory¹². This approach was also used to calculate the sea ice extent above stations of the observatory in relation to temporal dynamics of benthic megafauna¹³.

Other examples from other institutes involve, for example, the effect of the shift from multi-year to predominantly first year ice which travels faster¹⁴ and its effect on the exchange of any contaminated ice within the exclusive economic zones¹⁵.

However, to estimate the uncertainties of the calculated sea ice trajectories using satellite sea ice motion and concentration data, the pathways of 39 buoys were re-tracked. Buoy data were obtained from the SealcePortal.de and followed from their deployment position in a forward direction. Figure 1 shows the distance between buoys and virtual tracks over drift distance (upper panel) and time (lower panel). On average, the displacement of virtual buoys during the first 150 days (around 1000 km of ice drift) is around 35 km. After one year (ice drift of more than 2500 km), the average displacement is around 150 km. Given the low displacement we believe that the method is accurate enough to identify potential source areas and pathways of sea ice. For a detailed discussion of uncertainties associated to different motion products we refer to Sumata et al.^{16,17}.

Fig.1: Displacement between virtual and real drifting buoys in the Arctic.

The tool used at the AWI is called ICETrack and a detailed description of the applied method can be found in Krumpen 2017¹⁸

In the revised version we introduced this backtracking concept already in the introduction. L30

“A useful method to study sea ice drift pattern is by using passive microwave satellite images combined with the motions of sea ice buoys^{17,18}, which highlight the role of sea ice, e.g., for distributing oil spills¹⁹.”

Throughout the text we referred to the literature also mentioned by reviewer 3.

In the supplementary method section you find the error estimate L23:

“To quantify uncertainties of estimated sea ice trajectories using satellite sea ice motion and concentration data, pathways of 39 buoys were re-tracked. Buoy data were obtained from the SealcePortal.de and followed from their deployment position in a forward direction. On average, the

displacement of virtual buoys during the first 150 days (around 1000 km of ice drift) is around 35 km. After one year (ice drift of more than 2500 km), the average displacement is around 150 km.”

[3] Please use the same “unit” in measuring microplastic concentration throughout the paper. The unit “m⁻³ (P.2)”, “MP particles L-1 (P.4)”, “MP L-1 (P.4)”, “N/m³ (Fig. 1)”, and “N L-1 (Fig. 2)” are used for the microplastic concentrations, which make scientists uncomfortable. Please note, we differentiated between the entire cores, whose values were given in N m⁻³ (Fig. 1) to be able to upscale this data easily to larger volumes. For the individual sections of the core (Fig. 4) we used the unit N L⁻¹, the latter been used throughout the text in the revised version.

Specific points

[4] line 57, “one to two orders”; As mentioned in my comment [1], this should be “four orders”. Please see our answer to comment 1 above.

[5] line 100-101, “MP composition...sea ice floes”; The polymer types detected by the authors are mostly heavier than seawater (1.025 g/cm³) except PP and PE. Such “heavy” pelagic plastic fragments have been actually observed in upper Arctic Ocean? A recent paper by Cozar et al.⁴ showed significant amounts of floating plastic debris, particularly in the northernmost and easternmost areas of the Greenland and Barents seas. We therefore conclude that the described “heavy” pelagic plastic are present the Arctic. Moreover the MP polymers identified by Lusher et al.¹⁹ from surface waters south of Svalbard included polyester (15%), polyamide (15%), polyethylene (5%), acrylic (10%), polyvinyl chloride (5%) and cellulose (possibly Rayon, 30%). Of these, polyester, polyamide, acrylic and polyvinyl chloride (add up to 50% of MP) have a density higher than seawater and would be expected to sink. So, there are mechanisms at play in this region that keep these heavy MPs afloat.

[6] line 123, “originating from the Canadian basin”; In my eyes, it was originating from the boundary between Canadian and Nansen Basins (Fig. 2d).

We agree that the term Canadian basin is misleading in this sense and to avoid confusion, we changed this now to Amerasian basin, and also added the specific basin within the Amerasian basin. The new sentence L113 reads:

“This backtracking approach¹⁸ showed that the sea ice samples originated from different source areas, namely the Amerasian and Eurasian Basins (Fig. 1d; Tab. 1). In particular core B can be retraced to the Makarov Basin, while the cores within the Eurasian Basin originated from the Laptev Sea (core E) of Franz Josef Land (core D) and the deeper Nansen Basin (core C).”

[7] lines 154-155, “It can thus be assumed... Atlantic water inflow”; I could not understand how the authors assumed the origins of MPs. The similarities of “biological and environmental parameters (line 151)” between A,B,E and Transpolar Drift, and between C&D and Atlantic water inflow are required.

We agree that this was worded imprecisely, but given the newly computed data, we hope we made the statistical results clearer in this version.

[8] lines 214-215, “as those from South Korean waters...per volume unit”; These observations were conducted in an estuary and a lake, where microplastic concentrations are both completely different from those in open oceans such as the Arctic Ocean. Please refer Table 2 in Lusher et al. (2015), and

Table 1 in Isobe et al. (2017), both cited by the authors. The aim of this statement was to place our measured high numbers in context with the highest MP values published in LITTERBASE²⁰ to date to highlight the high level of pollution in this remote part of the world ocean. Despite replacing the lake study with a study from the Skagerrak, we decided to keep this sentence and did not refer to the suggested references by the reviewer, which were already discussed earlier in the MS. The new sentence reads L251:

“The core contained concentrations in the same league as those from South Korean waters⁴⁷ or the Skagerrak⁴⁸, which are the highest hitherto reported values in terms of measurements per volume unit. “

[9] lines 219-221, “The high PE concentrations...incoming Pacific inflow”; This is unacceptable speculation. It is unlikely to have a direct debris path from the “North Pacific Garbage Patch” within the subtropical gyre to the Arctic Ocean, passing through the subarctic gyre in the North Pacific. Although the authors may misunderstand that the “North Pacific Garbage Patch” is the unique hotspot of microplastics in the North Pacific, this is not the case; please see Table 1 of Isobe et al (2017).

We did not intend to create a vision of a debris path from the Pacific all the way into the Central Arctic; however, we need to explain the extremely high PE concentrations observed in core B from the Makarov Basin, with very unlikely local sources of this polymer. Since PE is neutrally buoyant, for us, it is a plausible idea to assume a transport of MP particles entering the Arctic also from the Pacific. The data in Isobe et al (Table 1)²¹ are based on Goldstein et al 2012²² and the presented value is the median of all samples studied. However, we would like to point out that the Goldstein study clearly states that there is an increase in MP particles of over two orders of magnitude in the North Pacific over the last decades. There is also extremely high particle MP loads in some of the samples, reaching particles of up to 32 m⁻³ with the 5th to 95th percentiles lying between 0.09 to 8.65 m⁻³ (suppl. Goldstein 2012²²). Furthermore, Desforges et al. 2014²³ also describe even higher MP particle loads of 279 m⁻³ for the NE Pacific. Given the general current patterns we think it is valid to speculate that the NE Pacific may be a source of MP in the Amerasian basin, especially since we clearly define that this is speculative. The new text reads as L255

“The Canadian Basin is supplied with water originating from the north-east Pacific and transported through the Bering Strait⁵⁰. From the southern part of the Chukchi Sea it has been modelled, based on drift trajectories of buoys that sea ice has a direct path from the Central Arctic towards the Fram Strait⁵¹. It might be therefore speculated that the high PE concentrations in the core from the Makarov Basin might reflect remains from the so called North Pacific Garbage Patch⁵², transported with the incoming Pacific inflow. A recent study by Desforges et al.⁵³ showed quite high MP concentrations for the NE Pacific and highlights the role of oceanographic conditions for the accumulation patterns of MP’s⁵³. Indeed, modelling indicates that on long time scales inter-ocean exchanges play a significant role in the distribution of marine debris enabling transport between accumulation areas⁵².”

[10] lines 332-335, “the mean value of ...per litre melted ice”; One of my major concerns in reading this paper is here. It seems likely that they extrapolated from “measured areas” to “total area”, and converted from “total area” to “particle numbers”. The extrapolation and conversion need both careful validation and error estimates. Also, the detailed procedure from “area” to “number” should be mentioned here. What I concerned is the serious overestimation of particle numbers in converting the area measured using the micro FTIR. As we partly shared the concerns highlighted by the reviewer, we performed the measurements as technical triplicates to reduce potential overestimation. As stated in L338

“Each horizon was weighed before being melted in glass-preserving jars at room temperature and then concentrated onto Anodisc filters (47 mm, Whatman, Freiburg, Germany). All samples were treated with of 35% H₂O₂ (Roth, Karlsruhe, Germany, filtered over 0.2 μm Anodisc). After filtering the melt water the filter was overlaid with 40 mL H₂O₂ and incubated at room temperature overnight. Lastly, the H₂O₂ was drained and the filters were flushed with approx. 750 mL MilliQ water.”

All meltwater from the section of ice cores was concentrated on the filter and afterwards treated with H₂O₂. We measured three measurement fields in different regions of the filter cake, as we could not measure the whole filter area within one measurement (see L363):

"As it was not possible to analyse the whole filter cake of (36 mm) in diameter in a single measurement, three separate fields on the concentrated sample where as technical triplicates measured."

Afterwards the mean value of the particle numbers of the three fields was calculated and scaled to the concentrated filter area. To show this process we adjusted the paragraph L367:

*“**Quantification and identification method:** Each measurement field was subjected to the automated analysis by Pimpke et al.²⁹. During this process each spectrum is compared twice against a spectral library with different data handling. Each successful hit was stored with together with the x,y and a quality factor into a csv file. Afterwards the file was analysed by image analysis determining the polymer types, particle number per polymer and size distribution¹. For each filter, the mean value N_F and standard deviation of the three m technical triplicates were calculated. To extrapolate to the total area in contact with the sample the following equation was used:*

$$N = \frac{N_F}{V_F}$$

The derived particle (N) numbers per litre melted ice were calculated from the mean value N_F and the volume fraction of one measurement field V_F from the overall volume. Particle numbers derived from blank samples were subtracted. from N_F. To estimate errors of this conversion we performed an error propagation which is provided in the supplement method section.”

[11] Figure S1(A and B); I am not familiar with nMDS analysis, and thus, please ignore this comment if this comes from my misunderstanding. Nonetheless, I have to state something that, in general, graphs without both the abscissa and ordinate are meaningless in sciences. nMDS plots are a common tool used in all molecular and many ecological studies. Such plots do not have axes, but the proximity of points indicates the degree of overlap or similarity of samples. However, we understand that people outside the field might not understand this and, therefore, only used the more intuitive cluster version of the statistical results S. Fig 2-5.

Reviewer #3 (Remarks to the Author):

Title: Arctic sea ice is an important temporal sink and means of transport for microplastic

Authors: Ilka Peeken, Sebastian Primpke, Birte Beyer, Julia Guetermann, Thomas Krumpfen, Melanie Bergmann, Laura Hehemann, Gunnar Gerds.

In this paper, the authors are reporting on microplastic loads in ice cores collected in the Arctic Ocean in the spring 2014 and summer 2015 Polarstern cruises. The data set is potentially interesting, but the paper is vague and the analysis of the data collected is not in depth.

The paper simply describes the results without telling the reader a story from the data.

This statement of the reviewer is in contrast to reviewer 1 and 2 who rather claim over interpretation of our results. In the current version, we changed the line of arguments substantially and also include several references suggested by the reviewer to put our data into wider context, hoping that the current story is more sound and that the reviewer is more satisfied with the revised version.

The paper also does not cite relevant literature nor present the results in the context of prior work.

We are sorry for not using all relevant literature and we thank the reviewer for his excellent addition of all the papers below which were very helpful to shape the story of the revision. Please see the annotated version of the original manuscript for the fundamentally revised version.

For these reasons, I recommend rejection of the paper in its present form.

Major Points:

1- The back trajectory model could have been used together with a simple 1D thermodynamic model to reconstruct the surface microplastic fields in the Arctic Ocean - as was done in Pfirman et al. (2004). This would add a new dimension and more depth to the paper.

This is a very good idea, however, we believe that no simple 1D thermodynamic model exists that reliably reconstructs melt and growth processes. We, therefore, refrain from including this model. To accurately describe thermodynamic growth we need, at least, initial thickness, surface temperature from sensible and latent heat fluxes, snow, etc., as has also been realized in the Pfirman et al. 2004²⁴ publication and which are still not available in the requested precision.

2- The Arctic is presented by the authors as a pristine environment; yet the fact that pollutants are present in the Arctic is a known fact for a while – see for instance AMAP publications Arctic Monitoring and Assessment Programme, Pfirman et al., (1997), Rigor and Colony (1997), Korsnes et al. (2002), Book, *ordered* Pavlov (2007).

We agree that the term pristine (line 203) was misleading, however, in our context it only referred to MP particles. We are aware that the Arctic as a whole is far from being pristine and we changed the text to L228 to: “free of plastic litter”. However, we thank the reviewer for the mentioned literature, which, overall, gave us new insights into some processes which support our story and have been incorporated in the revised MS. We further changed the introduction L27:

“It is well known that regions of the Arctic Ocean are highly polluted both due to local sources and long range atmospheric input¹³. In this context sea ice has been identified early on as a major vehicle of transport for various pollutants^{14, 15} with north and east Greenland as well as the Laptev Sea, being especially prone to contamination from several sources¹⁶. “

and the discussion L208:

“The role of sea ice to redistribute e.g. coastal sediments⁴⁰ as well as contaminants^{16, 41} along the Transpolar Drift or into the Central Arctic have long been recognised. It could be shown that the particular region of the Fram Strait will always be reached by any contamination source from the distant Arctic. Time between contamination and arrival in the Fram Strait has been calculated to be between two to four years for sources in the Laptev and Kara Sea, and up to six to eleven years from sources of the Amerasian Basin¹⁶. “

3- None of the Lagrangian Tracking literature is reviewed except for a recent paper by co-author Krumpfen et al.; see for instance, Tschudi et al 2010., Fowler et al. 2013), Newton et al 2017., Blanken et al. 2014, Szanyi et al (2016).

We thank the reviewer for pointing this out, although we only use this method as a common tool and, therefore, did not comment on this in the original version. However, since this was considered as a major point during the review process we included now in the introduction and discussion some of the above mentioned publications:

L30

“A useful method to study sea ice drift pattern is by using passive microwave satellite images combined with the motions of sea ice buoys^{17, 18}, which highlight the role of sea ice, e.g., for distributing oil spills¹⁹. Recent studies particularly highlight the changes involved with the shift to first year ice resulting in the tendency of sea ice floes to diverge from the main tracks²⁰ as e.g. the Transpolar Drift, with a complex effect of exchange processes of any contaminants between the exclusive economic zones (EEZ) of the various Arctic nations²¹. “

L300

“Given the climate change induced change to mainly first year ice in the Arctic, a recent study showed that most ice growing on the continental shelves has currently only a short travel path before it melts again and this would keep the contaminants in the regions of ice production²¹. “

Unfortunately, we could not refer to all of these publications due to the journal's limit of 70# references cited.

See also a more detailed answer to the backtracking approach in our reply made to Reviewer 1, comment 2.

4- The concept that microplastic can accumulate at the surface because there is generally more ice growth than melt in the Arctic is not discussed. For instance, microplastic incorporated in sea ice at the base when ice freezes will migrate upward year after year and can accumulate at the surface. This is not mentioned in the discussion of the results and the large concentration observed at the surface in some cores; Wollenburg, 1993; Nuernberg et al., 1994.

Thank you for pointing this out, however, the process described in the references only applies to Multi Year Ice (MYI) and unlikely to be very relevant in the currently changed Arctic, where large quantities of the MYI have been transformed to first year ice. However, as suggested by the reviewer, indeed it might explain our high values in the second year ice and we included this now in the text L244:

“An accumulation of sediment particles in the surface horizons, as has been described for multiyear ice⁴⁰, due to constant surface melting, might explain the high MP concentration observed in the surface of the

cores originating from the Makarov Basin and the Laptev Sea. In both cores the low salinity in the surface layers indicated the previous melting. Since we primarily observed low surface concentrations in the top of sea ice cores it is unlikely that this redistribution process occurs in in first year sea ice.”.

5- The authors say the microplastic are conservative in places and in others they say that it can be lost through brine channels.

Since currently, the dynamics of scavenging and releasing MP within and from sea ice are not well understood; we tried to take analogies from other processes which have been found in sea ice. Applying these processes to MP involves some speculation and we think we phrased this accordingly.

References:

1. Pripke S, Lorenz C, Rascher-Friesenhausen R, Gerdt G. An automated approach for microplastics analysis using focal plane array (FPA) FTIR microscopy and image analysis. *Analytical Methods* **9**, 1499-1511 (2017).
2. Löder MGJ, Gerdt G. Methodology used for the detection and identification of microplastics – a critical appraisal. In: *Marine Anthropogenic Litter*. (ed[^](eds Bergmann M, Gutow L., M. K). Springer (2015).
3. Tagg AS, Sapp M, Harrison JP, Ojeda JsJ. Identification and quantification of microplastics in wastewater using focal plane array-based reflectance micro-FT-IR imaging. *Analytical chemistry* **87**, 6032-6040 (2015).
4. Cózar A, *et al.* The Arctic Ocean as a dead end for floating plastics in the North Atlantic branch of the Thermohaline Circulation. *Science Advances* **3**, e1600582 (2017).
5. Obbard RW, Sadri S, Wong YQ, Khitun AA, Baker I, Thompson RC. Global warming releases microplastic legacy frozen in Arctic Sea ice. *Earth's Future* **2**, 315-320 (2014).
6. Nurnberg D, *et al.* Sediments in Arctic Sea-Ice - Implications for Entrainment, Transport and Release. *Mar Geol* **119**, 185-214 (1994).
7. Emery WJ, Fowler C, Maslanik J. Satellite remote sensing of ice motion. *Oceanographic Applications of Remote Sensing*, 367-379 (1995).
8. Fowler C, Emery W, Tschudi M. Polar Pathfinder Daily 25 km EASE-Grid Sea Ice Motion Vectors, version 2, Natl. Snow and Ice Data Cent, Boulder, Colo, (2013).
9. David C, Lange B, Krumpfen T, Schaafsma F, van Franeker JA, Flores H. Under-ice distribution of polar cod *Boreogadus saida*. *Polar Biology* **39**, 981-994 (2016).
10. Hardge K, Peeken I, Neuhaus S, Krumpfen T, Stoeck T, Metfies K. Sea Ice Origin and Sea Ice Retreat are Major Drivers of Variability in Arctic Marine Protist Composition. *Marine Ecology Progress Series*, (2017).

11. Krumpen T, *et al.* Recent summer sea ice thickness surveys in Fram Strait and associated ice volume fluxes. *The Cryosphere* **10**, 523 (2016).
12. Tekman MB, Krumpen T, Bergmann M. Marine litter on deep Arctic seafloor continues to increase and spreads to the North at the HAUSGARTEN observatory. *Deep-Sea Res Pt I* **120**, 88-99 (2017).
13. Taylor J, Krumpen T, Soltwedel T, Gutt J, Bergmann M. Regional-and local-scale variations in benthic megafaunal composition at the Arctic deep-sea observatory HAUSGARTEN. *Deep Sea Research Part I: Oceanographic Research Papers* **108**, 58-72 (2016).
14. Renner AHH, *et al.* Evidence of Arctic sea ice thinning from direct observations. *Geophys Res Lett* **41**, 5029-5036 (2014).
15. Newton R, Pfirman S, Tremblay B, DeRepentigny P. Increasing transnational sea-ice exchange in a changing Arctic Ocean. *Earths Future* **5**, 633-647 (2017).
16. Sumata H, Kwok R, Gerdes R, Kauker F, Karcher M. Uncertainty of Arctic summer ice drift assessed by high-resolution SAR data. *Journal of Geophysical Research: Oceans* **120**, 5285-5301 (2015).
17. Sumata H, *et al.* An intercomparison of Arctic ice drift products to deduce uncertainty estimates. *Journal of Geophysical Research: Oceans* **119**, 4887-4921 (2014).
18. Krumpen T. AWI ICETrack-Antarctic and Arctic Sea Ice Monitoring and Tracking Tool (Vers. 1.1). (ed[^](eds) (2017).
19. Lusher AL, Tirelli V, O'Connor I, Officer R. Microplastics in Arctic polar waters: the first reported values of particles in surface and sub-surface samples. *Sci Rep-Uk* **5**, (2015).
20. LITTERBASE 2017- Online Portal for Marine Litter. Alfred Wegener Institute Helmholtz Centre for Polar and Marine Research, Bremerhaven; www.litterbase.org
21. Isobe A, Uchiyama-Matsumoto K, Uchida K, Tokai T. Microplastics in the Southern Ocean. *Marine Pollution Bulletin*, (2016).
22. Goldstein MC, Rosenberg M, Cheng L. Increased oceanic microplastic debris enhances oviposition in an endemic pelagic insect. *Biol Letters* **8**, 817-820 (2012).
23. Desforges J-PW, Galbraith M, Dangerfield N, Ross PS. Widespread distribution of microplastics in subsurface seawater in the NE Pacific Ocean. *Marine pollution bulletin* **79**, 94-99 (2014).

24. Pfirman S, Haxby W, Eicken H, Jeffries M, Bauch D. Drifting Arctic sea ice archives changes in ocean surface conditions. *Geophys Res Lett* **31**, (2004).

Reviewers' comments:

Reviewer #1 (Remarks to the Author):

Thank you for addressing my comments,

I have read the manuscript and appreciate the large amount of work that has been done to edit the manuscript.

However I am still concerned about the following areas

Abstract

Line 11: The word source in this sentence still concerns me. I think this is a language issue rather a science one. When I think of source I expect it to mean where the microplastics have come from. I think what you mean is sample location: where you samples the cores from. I understand that the ice has moved and maybe why the word 'source' was used, but I find this confusing. Suggest you change 'source' to 'sample site'

Line 13: I disagree with this conclusion, your results show variation in microplastics observed, not local sources.

Introduction

Line 22: Although not specifically referred to at the initial review. I believe the 1% figure should be clarified, The Van Sebille paper cited specifically refers to small plastic items, and the manuscript needs to reflect this, and the fact of plastic has not been accounted for is because it is large items, on the seafloor or ingested (all of which have been documented but no global estimates have been made.

Line 42: 'change 'large' to 'largely'

Results

I appreciate there are no standardized methods for this field of science, but specially as your data show very large numbers of the smallest size class of plastics, would it not be helpful to use another measure rather than just number of microplastics when you are demonstrating abundance. One small piece of microplastic could, through the processing of the sample, become two smaller pieces, in this case doubling abundance. While this could be uniform across all samples, this is unlikely to be uniform across polymers (with some more brittle and more likely to fragment than others) therefore creating a bias. I am not suggesting that you need to reanalyse these data but comment on this should be clear made.

Methods

I am confused about the classification of varnish. I understand this to be a mix of resins and solvents each different depending on the product, therefore I am unsure how this could be classified in your results as this would mean comparing you data to multiple spectra especially as all other categories were single polymers

Reviewer #2 (Remarks to the Author):

Re-review of "Arctic sea ice is an important temporal sink and means of transport for microplastic" by Peeken et al.

I appreciate the authors addressing my previous concerns carefully. I am now pleased to recommend this paper for publishing in your journal. In particular, Figure 3 added in the revised manuscript is very impressive, and may change the observation standard for microplastics. The below are trivial comments which may make this paper more understandable.

- (1) P. 2, first paragraph: The unit "m-3" should be replaced with "N m-3" in line with Fig. 2.
- (2) P. 2, lines 66-67, "in a previous study of sea ice cores from th Central Arctic (1.3-9.6 x 10-4 m-3": I could understand the situation, and agree with the authors. However, the authors should describe how they computed these values (including the correction of "/m3" to "/L" in section 4 in Obbart et al. (2014)) in Methods or Supplement; otherwise the reader like me will be confusing.
- (3) P. 3, "Sea ice floe background and comparing to a previous study": Many geographical and ocean circulation names in the Arctic Ocean are revealed here, but only a part of them are shown in Figure 1. Thus, it is difficult to understand the descriptions regarding the paths of ice floes, especially for the readers unfamiliar with Arctic Ocean. Please add the geographical and/or ocean circulation names unless they are overcrowding in the figure.
- (4) References: Isobe et al (2016) should be updated to the latest citation (2017, 114(1), 623-626).

Reviewer #3 (Remarks to the Author):

The authors did address many of the concerns from the reviewers and the paper is more complete now. I disagree with the authors however that thermodynamic models are too uncertain to be used to backtrack the geographical location (spatial distribution) of the micro-plastic in the Arctic Ocean. There are melt layers in sea ice that provide constraints on the thermodynamic model and the initial thickness of the ice is known (from the core). I am surprised that Nature would contact me again for a second assessment of the paper after my first review and the response of the reviewer to my review.

My recommendation is still to reject the paper. The analysis remains incomplete, which makes the manuscript, in my opinion, below the high bar normally set for publication in Nature.

Answer to the reviewer

We are very thankful for the second evaluation of our work, which were very helpful to improve our manuscript. Please find below our detailed answer to the comments of the three reviewers.

Reviewers' comments:

Reviewer #1 (Remarks to the Author):

Thank you for addressing my comments,

I have read the manuscript and appreciate the large amount of work that has been done to edit the manuscript.

We thank the reviewer for acknowledging the changes we conducted for the resubmission of the manuscript and hope to be able to address the remaining concerns provided with the second review.

However I am still concerned about the following areas

Abstract

Line 11: The word source in this sentence still concerns me. I think this is a language issue rather a science one. When I think of source I expect it to mean where the microplastics have come from. I think what you mean is sample location: where you samples the cores from. I understand that the ice has moved and maybe why the word 'source' was used, but I find this confusing. Suggest you change 'source' to 'sample site'

We understand that this is misleading, but in the context we were referring to the source region of the growing sea ice, so sample site would not be the correct term. We omitted now the word “source area” and the new sentence line 10ff reads:

“Here, we show for the first time that sea ice MP has no uniform polymer composition, and that, depending on the growth region and drifting paths of the sea ice, unique MP patterns can be observed in different sea ice horizons.”

Line 13: I disagree with this conclusion, your results show variation in microplastics observed, not local sources.

For some MP species we can now clearly identify source regions (localized sources) after combining the backtracking with a 1-D thermodynamic sea ice growth model. This provides evidence, that we are not observing (random) variation in MP content, but actually different MP composition in different regions of the Arctic Ocean. The new sentence line 12ff reads:

“Thus even in remote regions such as the Arctic Ocean, certain MP particles indicate the presence of Thus even in remote regions such as the Arctic Ocean, certain MP indicate the presence of *localized* sources”

Introduction

Line 22: Although not specifically referred to at the initial review. I believe the 1% figure should be clarified, The Van Sebille paper cited specifically refers to small plastic items, and the manuscript needs to reflect this, and the fact of plastic has not been accounted for is because it is large items, on the seafloor or ingested (all of which have been documented but no global estimates have been made.

We agree this was misleading and changed the sentence line 20ff now too:

“However, only 1% of this has been accounted in terms of small plastic debris⁷, highlighting that some of the major sinks of oceanic plastic litter remains to be identified.”

Line 42: 'change 'large' to 'largely'

Done

Results

I appreciate there are no standardized methods for this field of science, but specially as your data show very large numbers of the smallest size class of plastics, would it not be helpful to use another measure rather than just number of microplastics when you are demonstrating abundance. One small piece of microplastic could, through the processing of the sample, become two smaller pieces, in this case doubling abundance. While this could be uniform across all samples, this is unlikely to be uniform across polymers (with some more brittle and more likely to fragment than others) therefore creating a bias. I am not suggesting that you need to reanalyse these data but comment on this should be clear made.

The reviewer raises a common discussion point in MP research and we completely agree with the concerns and therefore included the following sentence line 89ff

“We provide our measurements as particle count per volume for consistency with previous studies. However we suggest that future studies also consider polymer specific MP mass per volume data³⁴ to allow for calculation of fluxes or total load of synthetic polymers (independently of the degree of fragmentation). “

Methods

I am confused about the classification of varnish. I understand this to be a mix of resins and solvents each different depending on the product, therefore I am unsure how this could be classified in your results as this would mean comparing you data to multiple spectra especially as all other categories were single polymers

In our results each polymer type represents a group of several spectra of similar polymers which is necessary for the automated analysis. In total the database contains 270 spectra of different materials which were divided into 32 polymer types. For each polymer type as depicted in Figure 2 it was investigated if it was possible to differentiate between the individual spectra within the available spectral range. In the case of varnish it was found that it was not possible to determine the exact varnish but the chemical backbone based on polyurethanes or polyacrylates. Therefore all spectra of these three polymers had to be clustered into one polymer type. To improve the

clarity it was chosen to add additional group similar polymer types as highlighted in Figure 2 for PE and rubber. For the same reason we used varnish for the polymer type acrylates/PUR/varnish (as depicted in Figure 2).

To further highlight this we included the definition of the polymer type varnish in the results section (marked italic) line 93ff:

“In total seventeen different polymer types were identified (Fig. S1), with polyethylene (PE), varnish (*including polyurethanes and polyacrylates*), [...]”.

Reviewer #2 (Remarks to the Author):

Re-review of “Arctic sea ice is an important temporal sink and means of transport for microplastic” by Peeken et al.

I appreciate the authors addressing my previous concerns carefully. I am now pleased to recommend this paper for publishing in your journal. In particular, Figure 3 added in the revised manuscript is very impressive, and may change the observation standard for microplastics. The below are trivial comments which may make this paper more understandable.

We are very grateful for this positive evaluation of the manuscript.

(1) P. 2, first paragraph: The unit “m⁻³” should be replaced with “N m⁻³” in line with Fig. 2.

Done.

(2) P. 2, lines 66-67, “in a previous study of sea ice cores from the Central Arctic (1.3-9.6 x 10⁻⁴ m⁻³”: I could understand the situation, and agree with the authors. However, the authors should describe how they computed these values (including the correction of “/m³” to “/L” in section 4 in Obbard et al. (2014)) in Methods or Supplement; otherwise the reader like me will be confusing.

We followed the suggestion of the reviewer and included now a paragraph in the method section, line 388ff

“To compare ours with the previous study of MP in sea ice, we also excluded rayon from the MP identified in Obbard et al. 2014². We digitized the data from Figure 2 (which according to the erratum are given in “per liter”) and up-scaled the numbers to N m⁻³ for Fig. 1b.

In addition we refer to this in the sentence line 65ff (marked italic):

“The values recorded in this study are two to three orders of magnitude higher than in a previous study from the Central Arctic² (1.3 -9.6 × 10⁴ N m⁻³, values exclude Rayon, *for further details see method section*), which can largely be explained by the different methodology used.”

(3) P. 3, “Sea ice floe background and comparing to a previous study”: Many geographical and ocean circulation names in the Arctic Ocean are revealed here, but only a part of them are shown in Figure 1. Thus, it is difficult to understand the descriptions regarding the paths of ice floes, especially for the readers unfamiliar with Arctic Ocean. Please add the geographical and/or ocean circulation names unless they are overcrowding in the figure.

We included now all geographical names which are mentioned in the text both in Figure 1a but explicitly in Fig. 1d to make it easier for the reader to understand this section.

(4) References: Isobe et al (2016) should be updated to the latest citation (2017, 114(1), 623-626).

Done.

Reviewer #3 (Remarks to the Author):

The authors did address many of the concerns from the reviewers and the paper is more complete now. I disagree with the authors however that thermodynamic models are too uncertain to be used to backtrack the geographical location (spatial distribution) of the micro-plastic in the Arctic Ocean. There are melt layers in sea ice that provide constraints on the thermodynamic model and the initial thickness of the ice is known (from the core). I am surprised that Nature would contact me again for a second assessment of the paper after my first review and the response of the reviewer to my review.

Even though we are still convinced, that one-dimensional modeling of backtracked ice floes comes with large uncertainties, we agree with reviewer 3, that such a model can be used to support the claims we are making regarding source areas of MP. We thus incorporated the one dimensional thermodynamic sea-ice model suggested by reviewer 3 to link the different samples to a geographic origin. As expected this supports our interpretations regarding the source regions of micro plastics. Due to its nature of supporting interpretation, we decided to include respective text and figures to the supplementary material and refer to it in the main text.

My recommendation is still to reject the paper. The analysis remains incomplete, which makes the manuscript, in my opinion, below the high bar normally set for publication in Nature.

By including the one dimensional thermodynamic model exactly as suggested by reviewer 3, we hope to have successfully addressed all reviewer concerns.

REVIEWERS' COMMENTS:

Reviewer #3 (Remarks to the Author):

The authors have addressed my outstanding comment satisfactorily.

I recommend that the paper be accepted for publication in its present form.

Answer to the reviewer

We are very thankful for the third evaluation of our work. Please find below our answer to the comments of reviewer 3.

Reviewer #3 (Remarks to the Author):

The authors have addressed my outstanding comment satisfactorily.

I recommend that the paper be accepted for publication in its present form.

We are very pleased to be able to address the concern of reviewer 3 to his expectations.